# Odd-paired is a pioneer-like factor that coordinates with Zelda to control gene expression in embryos

Theodora Koromila[1], Fan Gao[1], Yasuno Iwasaki[2], Peng He[1], Lior Pachter[1], J Peter Gergen[2], Angelike Stathopoulos[1]*

[1]California Institute of Technology, Division of Biology and Biological Engineering, Pasadena, United States; [2]Stony Brook University, Department of Biochemistry and Cell Biology and Center for Developmental Genetics, Stony Brook, United States

**Abstract** Pioneer factors such as Zelda (Zld) help initiate zygotic transcription in *Drosophila* early embryos, but whether other factors support this dynamic process is unclear. Odd-paired (Opa), a zinc-finger transcription factor expressed at cellularization, controls the transition of genes from pair-rule to segmental patterns along the anterior-posterior axis. Finding that Opa also regulates expression through enhancer *sog_Distal* along the dorso-ventral axis, we hypothesized Opa's role is more general. Chromatin-immunoprecipitation (ChIP-seq) confirmed its in vivo binding to *sog_Distal* but also identified widespread binding throughout the genome, comparable to Zld. Furthermore, chromatin assays (ATAC-seq) demonstrate that Opa, like Zld, influences chromatin accessibility genome-wide at cellularization, suggesting both are pioneer factors with common as well as distinct targets. Lastly, embryos lacking *opa* exhibit widespread, late patterning defects spanning both axes. Collectively, these data suggest Opa is a general timing factor and likely late-acting pioneer factor that drives a secondary wave of zygotic gene expression.

*For correspondence:
angelike@caltech.edu

**Competing interests:** The authors declare that no competing interests exist.

## Introduction

The transition from dependence on maternal transcripts deposited into the egg to newly transcribed zygotic transcripts is carefully regulated to ensure proper development of early embryos. During the maternal-to-zygotic transition (MZT), maternal products are cleared and zygotic genome activation occurs (rev. in *Vastenhouw et al., 2019*; *Hamm and Harrison, 2018*). In *Drosophila* embryos, the first 13 mitotic divisions involve rapid nuclear cycles (nc), that include only a short DNA replication S phase and no G2 phase, and the nuclei are not enclosed in separate membrane compartments but instead present in a shared cytoplasm (*Foe and Alberts, 1983*). This streamlined division cycle likely relates to the fast development of *Drosophila* embryos, permitting rapid increase in cell number before gastrulation in a matter of a few hours. Gene expression is initiated during the early syncytial stage, as early as nc7, and continues to the cellularized blastoderm stage (*Ali-Murthy and Kornberg, 2016*; *Lott et al., 2011*; *Kwasnieski et al., 2019*). Gene expression patterns may be transient or continuous, lasting through gastrulation or beyond (*Kvon et al., 2014*). This process is controlled by a specific class of transcription factors called pioneer factors, which bind to closed chromatin cis-regulatory regions to create accessible binding sites for additional transcription factors during development (*Iwafuchi-Doi and Zaret, 2014*). The pioneer Zelda (Zld) is a ubiquitous, maternal factor that binds to promoters of the earliest zygotically expressed genes and primes them for activation (*Liang et al., 2008*; *Harrison et al., 2010*; *Harrison et al., 2011*). It was unknown whether a similar regulation exists in other animals until the zebrafish Pou5f1, homolog of mammalian Oct4, was shown to act in an analogous manner to Zld in that it controls zygotic gene activation in vertebrates (*Leichsenring et al., 2013*).

A complete understanding of how widespread activation of zygotic gene expression is achieved is lacking, but several regulatory mechanisms have been proposed. One model suggests that a decrease in histone levels over time due to dilution during nuclear division provides an opportunity for the pioneer factors that drive zygotic gene expression to successfully compete for DNA access and activate transcription (*Shindo and Amodeo, 2019*; *Hamm and Harrison, 2018*). Chromatin accessibility can also be more specifically modulated by targeted action of transcriptional factors at regulatory loci. For example, Zld is pivotal for the MZT as it increases accessibility of chromatin at enhancers thereby allowing binding of other transcriptional activators at these DNA regions which facilitates initiation of zygotic gene expression (*Xu et al., 2014*; *Harrison et al., 2011*; *Liang et al., 2008*; *Nien et al., 2011*; *Yáñez-Cuna et al., 2012*). Zld binds nucleosomes, another characteristic of pioneer factors (*McDaniel et al., 2019*), and therefore loss of Zld leads to a global decrease in zygotic gene expression as many enhancer regions remain inaccessible (*Schulz et al., 2015*; *Sun et al., 2015*). Through its effects on chromatin accessibility, Zld has been shown to influence the ability of morphogen transcription factors, Bicoid and Dorsal, to support embryonic patterning (*Xu et al., 2014*; *Foo et al., 2014*). While Zld is clearly pivotal for supporting MZT, some genes continue to be expressed even in its absence (*Nien et al., 2011*). As chromatin accessibility in the early embryo has recently been shown to be a dynamic process (*Blythe and Wieschaus, 2016b*; *Bozek et al., 2019*), it is possible that Zld contributes in a stage-specific manner and that other as yet unidentified pioneer factors contribute to the extended process of zygotic genome activation.

The embryo undergoes a widespread state change after the 14th nuclear division, termed the midblastula transition (MBT) (*Foe and Alberts, 1983*; *Shermoen et al., 2010*). This developmental milestone is marked by dramatic slowing of the division cycle and cellularization of nuclei before the onset of embryonic programs of morphogenesis and differentiation. Cell membranes encapsulate nuclei to form a single-layered epithelium. In addition, at nc14, developmental changes relating to DNA replication occur; namely a lengthened S-phase and the introduction of G2 phase into the cell cycle. MBT is also associated with clearance of a subset of maternally provided mRNAs, large-scale transcriptional activation of the zygotic genome, and an increase in cell cycle length (*Yuan et al., 2016*; *Tadros and Lipshitz, 2009*). We hypothesized that other late-acting pioneer factors manage the MBT in addition to or in place of Zld.

The *Drosophila* gene *odd-paired* (*opa*) encodes the founding member of the Zinc finger in the cerebellum (Zic) protein family (*Aruga et al., 1996*; *Hursh and Stultz, 2018*). The important regulatory role of Zic (ZIC human ortholog) in early developmental processes has been established across major animal models and also implicated in human pathology (rev. in *Aruga and Millen, 2018*; *Houtmeyers et al., 2013*). *opa* is a broadly expressed gene of relatively long transcript length (~17 kB) that is activated during mid-nc14 and serves a number of important functions throughout development (*Cimbora and Sakonju, 1995*; *Benedyk et al., 1994*). Opa protein has a DNA-binding domain containing five Cys2His2-type zinc fingers, and shares homology with mammalian Zic1, 2, and three transcription factors. While mutants exhibit a pair-rule phenotype (*Jürgens et al., 1984*), the broad expression pattern of *opa* contrasts with the typical 7-stripe pattern of other pair-rule genes. Rather than providing spatial information as do most other pair-rule transcription factors, Opa instead acts as a timing factor to broadly regulate the expression of segment polarity genes including the transition of pair-rule genes to segmental expression patterns (i.e. from 7- to 14-stripes) (*Clark and Akam, 2016*; *Benedyk et al., 1994*). *opa* mutant embryos die before hatching and in addition to aberrant segmentation, they also exhibit defects in larval midgut formation (*Cimbora and Sakonju, 1995*). During midgut formation, Opa regulates expression of a pivotal receptor tyrosine kinase required for proper morphogenesis of the visceral mesoderm (*Mendoza-García et al., 2017*). In addition, at later stages, Opa supports temporal patterning of intermediate neural progenitors of the *Drosophila* larval brain (*Abdusselamoglu et al., 2019*).

Previous studies suggested that Opa can influence the activity of other transcription factors to promote gene expression. A well-characterized target of Opa in the early embryo is *sloppy-paired 1* (*slp1*), a gene exhibiting a segment polarity expression pattern and for which two distinct enhancers have been identified that are capable of responding to regulation by Opa and other pair-rule transcription factors including Runt (Run; *Cadigan et al., 1994*; *Prazak et al., 2010*). One of these, the *slp1* DESE enhancer, mediates both Run-dependent repression and activation and Opa plays a central role by supporting Run's activating input (*Hang and Gergen, 2017*). Additionally, our recent study showed that Run regulates the spatiotemporal response of another enhancer, *sog_Distal*

(*Ozdemir et al., 2011*; also known as *sog_Shadow*; *Hong et al., 2008*) to support its expression in a broad stripe across the dorsal-ventral (DV) axis on both sides of the embryo (*Koromila and Statho-poulos, 2019*). Using a combination of fixed and live imaging approaches, our analysis suggested that Run's role changes from repressor to activator over time in the context of *sog_Distal*; late expression requires Run activating input. These analyses of *slp1* DESE and *sog_Distal* regulation support the view that Opa might provide temporal input at enhancers.

The current study was initiated to investigate whether Opa supports late expression through the *sog_Distal* enhancer. Previous studies had not linked Opa to the regulation of DV patterning. Nevertheless, through mutagenesis experiments coupled with in vivo imaging, we provide evidence that Opa does regulate expression of the *sog_Distal* enhancer. Further, we show that Opa's role is indeed late-acting, occurring in embryos at mid-nc14 onwards, whereas the enhancer initiates expression at nc10. Given its ability to regulate key embryonic enhancers in a temporal manner, we hypothesized that Opa may play a general role in activating zygotic gene expression during late MZT, much as Zld does earlier. To assay Opa's genome-wide effects on gene expression and chromatin accessibility in the embryo, we used a combination of sequencing approaches: RNA-seq transcriptome profiling, chromatin immunoprecipitation (ChIP-seq) and single-embryo Assay for Transposase-Accessible Chromatin (ATAC-seq). Our whole-genome data demonstrate that Opa contributes to patterning the embryo by serving as a general timing factor, and possibly as a pioneer, to broadly influence zygotic transcription in nc14, as the embryo undergoes cellularization, during late phase of the maternal-to-zygotic transition.

## Results

### Opa regulates the *sog_Distal* enhancer demonstrating a role for this gene in DV axis patterning

In a previous study, we created a reporter in which the 650 bp *sog_Distal* enhancer sequence was placed upstream of a heterologous promoter from the *even skipped* gene (*eve.p*), driving expression of a compound reporter gene containing both a tandem array of MS2 sites and the gene *yellow*, including its introns (*Koromila and Stathopoulos, 2017*). We used this reporter to assay gene expression supported by the *sog_Distal* enhancer in the early embryo. While this enhancer becomes active at nc10 and continues into gastrulation, in this study we focused on late expression through *sog_Distal* during nc13 and nc14. Due to its length (i.e. ~45 min compared to ~15 min for nc13 at 23°C) nc14 was assayed in four, roughly 12 min intervals: nc14A, nc14B, nc14C, and nc14D. Live movies were analyzed using a previously defined computational approach tailored to spatiotemporal dynamics (*Koromila and Stathopoulos, 2019*).

In our previous study, mutation of the single Run binding site in the *sog_Distal* enhancer led to expansion of reporter expression early (i.e. nc13 and early nc14) but loss of expression late (i.e. nc14C and nc14D) (*Koromila and Stathopoulos, 2019*). These results suggested that Run's role switches from that of repressor to activator in the context of *sog_Distal* enhancer during this time. Other studies also suggested that Run can function as either repressor or activator depending on context, as the response of a given enhancer to Run is influenced by the presence or absence of other transcription factors (*Hang and Gergen, 2017*; *Prazak et al., 2010*; *Swantek and Gergen, 2004*). This is the case for *slp1*, where Opa is required for Run-dependent activation of expression (*Swantek and Gergen, 2004*). We therefore hypothesized that Opa might also influence Run's activity with respect to the *sog_Distal* enhancer; specifically, that Opa functions to support late expression of *sog_Distal*, when Run switches to providing activating input (*Koromila and Stathopoulos, 2019*).

In concordance with this hypothesis, the *sog Distal* 650 bp enhancer sequence contains five putative 12 bp Opa binding sites, based on comparison with the vertebrate Zic3 consensus motif (JAS-PAR; *Figure 1I*). We introduced 2–4 bp mutations at these five sites (i.e. *sogD_ΔOpa*) and assayed MS2-MCP reporter expression by in vivo imaging of nascent transcription (*Garcia et al., 2013*; *Lucas et al., 2013*). We found that expression was relatively normal up to stage nc14B but then exhibited a visually apparent decrease at nc14C (*Figure 1C* compare to *Figure 1A*; *Video 1*). Quantitative analysis of MS2-MCP signal in embryos containing either the wildtype *sog_Distal* or *sogD_ΔOpa* reporters using a previously described analysis pipeline (*Koromila and Stathopoulos,*

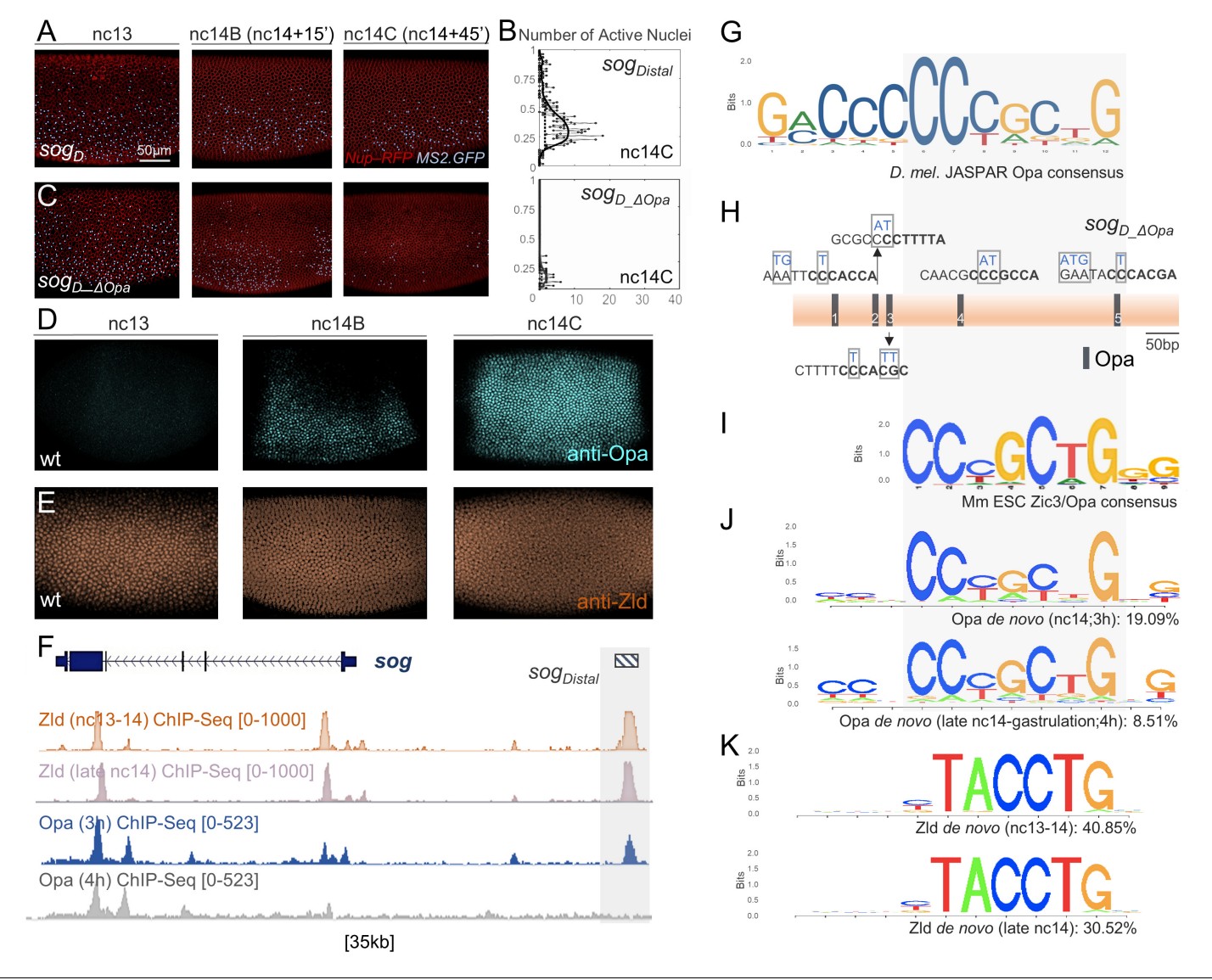

**Figure 1.** Opa is required to support activation of reporter expression at late nc14, just preceding gastrulation. In this and all other subsequent figures lateral views of embryos are shown with anterior to the left and dorsal up, unless otherwise noted. (**A,C**) Stills from movies (n = 3 for each) of the two indicated *sog_Distal* MS2-yellow reporter variants *sog_Distal* (**A**) or *sogD_ΔOpa* (**C**) in which five predicted Opa-binding sites were mutated as shown (**H**) and transcription detected in vivo via MS2-MCP-GFP imaging (*Koromila and Stathopoulos, 2019*) at three representative timepoints: nc13, nc14B, and nc14C. Blue dots indicate presence of GFP+ signal, representing nascent transcripts labeled by the MS2-MCP system; thresholding was applied and remaining signals identified by the Imaris Bitplane software, for visualization purposes only. Nuclei were labeled by Nup-RFP (*Lucas et al., 2013*). Scale bar represents 50 μm. (**B**) Plots of number of active nuclei, defined by counting dots (x-axis) versus relative DV axis embryo-width (EW) position (y-axis), analyzed from representative stills from movies of three embryos at nc14C. (**D, E**) Anti-Opa (**D**) and anti-Zld (**E**) antibody staining of early wild-type embryos at the indicated stages. (**F**) Integrative Genomics Viewer (IGV) genome browser track of the *sog* locus showing Zld and Opa ChIP-seq data for embryos at two timepoints: nc13-14 and nc14 late for Zld (GSM763061 and GSM763061, respectively; *Harrison et al., 2011*) and 3 hr and 4 hr for Opa. Zld nc13-14, Zld nc14 late and Opa 3 hr ChIP-seq samples are of overlapping timepoints, whereas Opa 4 hr ChIP-seq sample is later. Gray shading marks the region of *sog_Distal* enhancer location. (**G**) JASPAR consensus binding site for Opa based on mammalian Zic proteins identified by bacterial one-hybrid (*Sen et al., 2010*; *Noyes et al., 2008*). (**H**) Location of 5 sequences within the 650 bp *sog_Distal* enhancer region that match the Jaspar Opa consensus binding site allowing 1 bp mismatch. Mutated Opa sites introduced to eliminate binding are shown in blue, creating *sogD_ΔOpa* (**C**; see Materials and methods). Bases in bold (7 bp) indicate matches to the Opa de novo motifs identified by ChIP-seq analysis (see **J**). For sake of comparison to consensus sequence, reverse complement sequence is shown for a subset. (**I**) Consensus binding site for *Mus musculus* Zic3/ Opa homolog identified using ChIP-seq (*Lim et al., 2010*). (**J,K**) Sequence logo representations of the most significant and abundant motifs, likely consensus binding sites, identified by HOMER de novo motif analysis in the Opa 3 hr and Opa 4 hr (**J**), or Zld nc13-14 and Zld nc14 late (**K**) ChIP-seq

*Figure 1 continued on next page*

*Figure 1 continued*

datasets defined (Central motif enrichment p-values 1e-566, 1e-354, 1e-3283, and 1e-2173, respectively). Grey-shaded box indicates the shared region between Opa motifs.

The online version of this article includes the following figure supplement(s) for figure 1:

**Figure supplement 1.** Assay of *sog_Distal* expression outputs through live in vivo imaging following mutagenesis of Opa or Run predicted binding sites.

**Figure supplement 2.** Most abundant motifs identified using HOMER de novo motif analysis within the Opa (3 hr) ChIP-seq and two Zld ChIP-seq datasets spanning nc14.

*2019*) confirms that *sog_Distal* expression is greatly reduced at nc14C for the mutant reporter compared to wildtype (*Figure 1B*). A similar loss of late expression only (i.e. nc14C onwards) was observed when even a single Opa site is mutated (*Figure 1—figure supplement 1B, B', E*) and this decrease is comparable to when the Run site is mutated (*Figure 1—figure supplement 1A, D*; *Koromila and Stathopoulos, 2019*). These results support the view that Opa promotes expression through *sog_Distal* from nc14C onwards, possibly, by supporting Run's switch from repressor to activator (see Discussion).

The timing of Opa expression supports a role for this factor in driving expression of *sog_Distal* at mid-nc14, approximately at the time of the MBT. Using an anti-Opa antibody (*Mendoza-García et al., 2017*), we examined spatiotemporal dynamics associated with Opa protein in the early embryo through analysis of localization in a time series of fixed embryos. Opa expression is absent at nc13, first observed at nc14B, and achieves its mature pattern approximately by nc14C (*Figure 1D*). The timing of Opa onset of expression correlates with the timing of loss of late expression from the *sog_Distal* reporter observed when Opa sites are mutated (i.e. *sogD_ΔOpa*; *Figure 1D*, compare with 1C). On the other hand, the ubiquitous, maternal transcription factor Zld is detected throughout this time period including during nc13 (*Figure 1E*). Loss of Zld input to *sog_Distal* through mutation of Zld binding sites leads to spatial retraction of the reporter pattern (*sog_Shadow*; *Yamada et al., 2019*) rather than an overall loss of expression as observed when Opa-binding sites are mutated (*Figure 1C*). Care was taken to preserve Zld and Run binding sites (and those of other predicted inputs: Dorsal, Twist, or Snail; *Figure 1—figure supplement 1*; *Koromila and Stathopoulos, 2019*) during generation of the Opa site mutant *sog_Distal* reporter (*Figure 1—figure supplement 1C*).

These results suggest that Opa regulates late expression of *sog_Distal*, specifically, from mid-nc14 onwards. Recent studies have demonstrated that the *opa* gene is generally important for the temporal regulation of anterior-posterior (AP) axis segmental patterning in *Drosophila* as well as in *Tribolium* (*Clark and Peel, 2018*; *Clark and Akam, 2016*). However, as our results suggest a role for Opa in the regulation of *sog* expression, which relates to DV patterning, we hypothesized Opa's role extends beyond control of segmentation to patterning of the embryo, in general.

## Use of anti-Opa antibody to conduct assay of in vivo genome occupancy through ChIP-seq analysis

To examine the in vivo DNA occupancy of Opa in early *Drosophila* embryos, we conducted

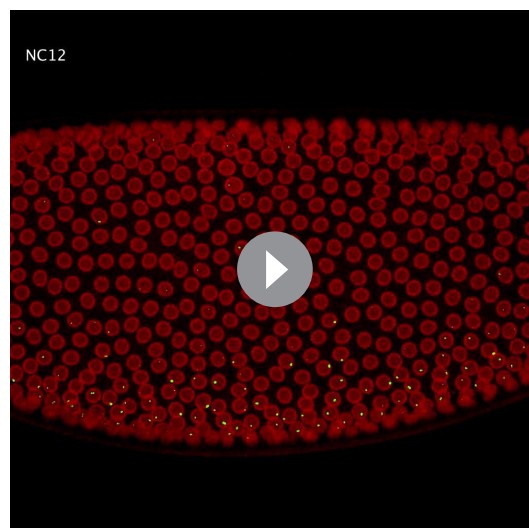

**Video 1.** Visualization of *sogD_ΔOpa* transcriptional activities in a representative early embryo from nc12 to gastrulation using MS2-MCP in vivo imaging. Expression normally extends until gastrulation for the wildtype reporter (*Koromila and Stathopoulos, 2019*) but upon mutation of five Opa-binding sites expression is extinguished by nc14C. Stills from the movie are shown in *Figure 1C*.
https://elifesciences.org/articles/59610#video1

chromatin immunoprecipitation coupled to high-throughput sequencing (ChIP-seq). Two different anti-Opa rabbit polyclonal antibodies were used to immunoprecipitate chromatin obtained from two embryo samples of average age 3 hr (roughly stages 5–6, encompassing nc14) or 4 hr (roughly stages 6–8, later than nc14) (see Methods). For the 3 hr ChIP-seq dataset, the MACS2 peak caller was used to identify 16,085 peaks, providing an estimate of the number of genomic positions occupied by Opa in vivo at this developmental timepoint. 200 bp regions centered at these peaks were analyzed using the HOMER program (*Heinz et al., 2010*) to identify overrepresented sequences that align to transcription factor binding motifs (see Materials and methods). The most significant hit, present in over 19% of all peaks, is a 7 bp core sequence with homology to the 12 bp Opa JASPAR consensus (*Figure 1J*, compare with 1G; *Figure 1—figure supplement 2*) as well as to mammalian homolog Zic transcription factors (e.g. see *Figure 1I*; *Lim et al., 2010*). A second motif exhibiting extended homology with the JASPAR Opa consensus was also identified through analysis of the Opa 3 hr ChIP-seq dataset, but this extended site is present at lower abundance (*Figure 1—figure supplement 2A, D*). In the *sog_Distal* enhancer sequence, the five Opa sites initially identified by comparison to the JASPAR motif also match the ChIP-seq-derived Opa de novo consensus in 6 of the 7 bases (*Figure 1H*; compare to 1J). However, there is a notable mismatch in the 3'-most position; while the de novo Opa consensus from the 3 hr ChIP-seq dataset does not include Adenine at this position, both the JASPAR site and de novo Opa consensus derived from the 4 hr ChIP-seq dataset do [*Figure 1H,J* (bottom motif)]. These sequence discrepancies may relate to differences in optimal affinities for binding sites within *sog_Distal* compared to those identified by ChIP-seq or they may indicate binding preferences dictated by the presence of heterodimeric binding partners.

We hypothesized that Opa might also regulate expression of *sog_Distal* late, following mid-nc14. Independent chromatin immunoprecipitation experiments have assayed Zld in vivo binding at two stages (i.e. nc13-nc14 and nc14 late; roughly equivalent to Opa 3 hr) and detected widespread binding of Zld throughout the genome including at *sog_Distal* (*Figure 1F*, top) (*Foo et al., 2014*; *Harrison et al., 2011*). Opa ChIP-seq detects Opa occupancy at *sog_Distal* during the 3 hr but not the 4 hr window (*Figure 1F*). At gastrulation, *sog_Distal* reporter expression changes from a broad lateral stripe to a thin stripe along the midline; it is possible that at this stage, expression of *sog_Distal* is no longer directly dependent on Opa.

Opa (3 hr) and Opa (4 hr) ChIP-seq experiments each identified ~16K peaks of occupancy representing locations in the genome that are occupied by Opa, with 9995 peaks in common (*Figure 2—figure supplement 1A*). This suggests that ~6K peaks are occupied early-only (i.e. nc14; including late nc14 when enhancers associated with segmentation are active) and a roughly equal number are occupied late-only (following gastrulation, stage 6–8) possibly relating to Opa's transition to a role in supporting visceral mesoderm specification (*Mendoza-García et al., 2017*) or other roles. Here, we focused on understanding Opa's initial actions during nc14; therefore, region overlap analysis was used to identify common regions of occupancy for Opa and Zld, using several independently obtained ChIP-seq datasets inclusive of nc14: Opa (3 hr) and both Zld (nc13-14) and Zld (nc14 late) (this study and *Harrison et al., 2011*, respectively) (see Materials and methods). Zld motifs derived de novo from the two Zld ChIP-seq datasets are almost identical (*Figure 1K*); however, the two datasets differ with respect to the most enriched de novo motifs identified for other factors (*Figure 1—figure supplement 2E, F*).

## Assay of overrepresented sites associated with Opa ChIP-seq peaks

The HOMER sequence analysis program was used to identify overrepresented motifs within the Opa (3 hr), Zld (nc13-14) and Zld (nc14 late) peaks as well as for three classes of peaks: Opa-only, Zld-only, or Opa-Zld overlap; in order to identify associated motifs that might provide insight into the differential or combined functions of Opa and Zld.

For the Opa 3 hr and Zld nc13-14 comparison, these datasets have 6087 peaks in common (Opa-Zld overlap), whereas 9998 regions were bound by Opa alone (Opa-only) and 10781 regions were bound by Zld alone (Zld-only) (*Figure 2A*). As expected, the top motifs in each class matched the Opa or Zld consensus sequences with 16% of the Opa-only peaks containing at least one Opa motif; 55% of the Zld-only peaks containing at least one Zld motif; and 5% and 15% of the Opa-Zld overlap peaks containing at least one Opa or one Zld motif, respectively (*Figure 2B–D*; see also *Figure 2—source data 1*). The second-most significant motif identified in each class of called peaks corresponds to Dref (6%) for Opa-only; Caudal (Cad; 14%) for Zld-only; and Trl/GAF (11%) for Opa-Zld

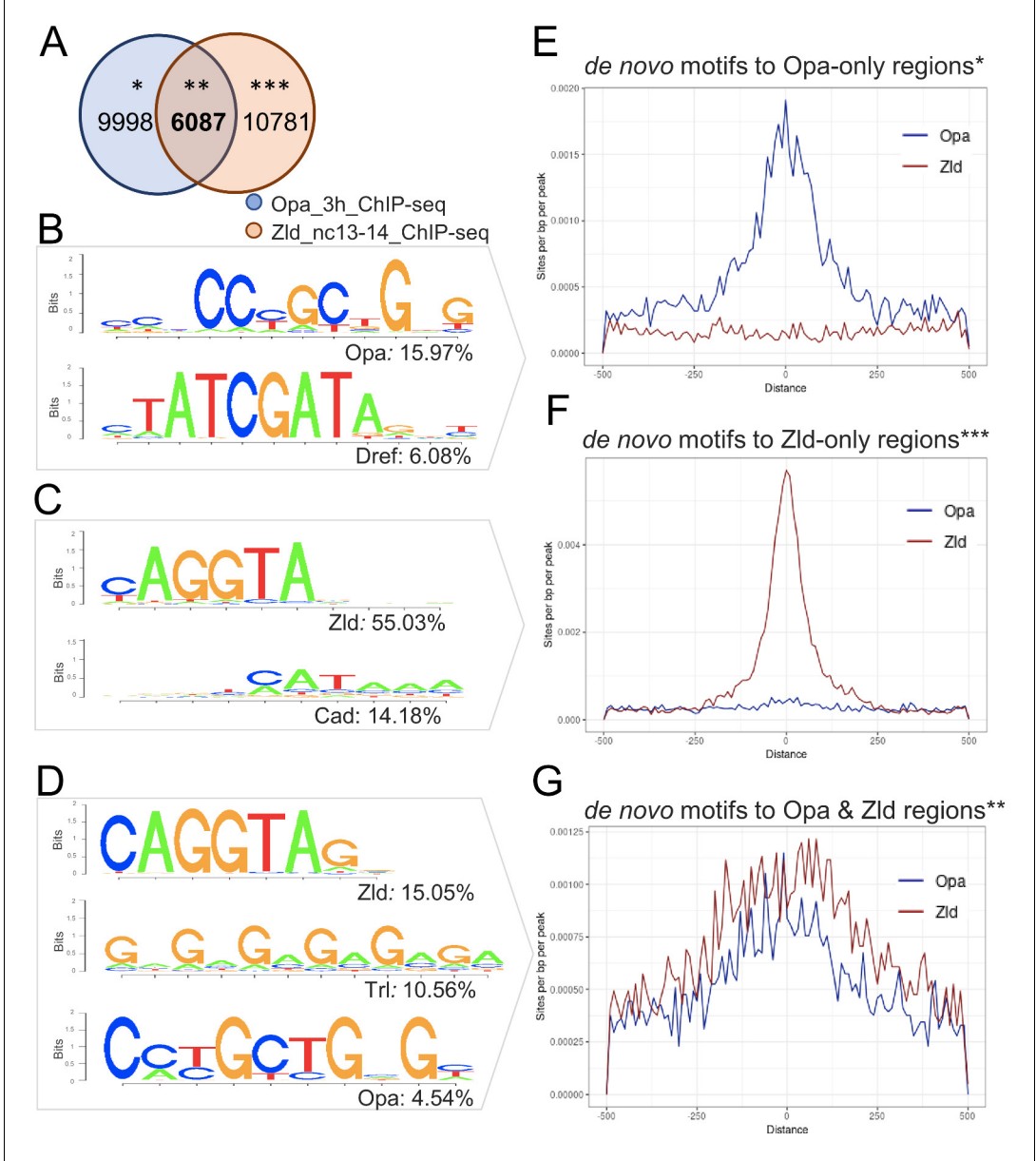

**Figure 2.** Enrichment of Opa and Zld de novo motifs in a subset of peaks that correspond to Opa-only, Zld-only, or Opa/Zld-bound regions identified by ChIP-seq. (A) Venn diagram showing overlap between peaks called using MACS2 analysis of Opa (3 hr) and Zld (nc13-14) ChIP-seq data. Opa and Zld experiments used embryos of 2.5–3.5 hr in age or nc13-14, respectively, which are overlapping timepoints. Opa-only peaks (*); Opa/Zld overlap peaks (**); Zld-only peaks (***). (B–D) Sequence logo representations of two to three most abundant motifs identified using HOMER de novo motif analysis within three sets of peaks: Opa-only, Zld-only, Opa/Zld overlap peaks (D). Sequence logo height indicates nucleotide frequency; corresponding percentage of peaks containing match to motifs also shown for each set, as indicated. p-Values represent the significance of motifs' enrichment compared with the genomic background, which is greater than 1e-43 in all cases. See also *Figure 2—source data 1*. (E–G) Aggregation plots showing enrichment of Opa or Zld de novo motifs identified within Opa-only, Zld-only, or Opa/Zld-bound regions from Opa (3 hr) peaks and Zld (nc13-nc14) ChIP-seq peaks. Averaging of ChIP-seq data from two replicates was performed prior to the de novo analysis. (E) Opa-only bound regions (after exclusion of Zld-only and Opa-Zld overlap peaks); (F) Zld-only bound regions (after exclusion of Opa and Opa-Zld overlap peaks); and (G) for Opa-Zld overlap regions.

The online version of this article includes the following source data and figure supplement(s) for figure 2:

**Source data 1.** Significance and abundance of motifs for known transcription factors found by HOMER within the three corresponding sets of peaks.

**Figure supplement 1.** Overlapping peaks identified in the Opa (3 hr), Opa (4 hr), and/or Zld (nc14 late) ChIP-seq datasets, as well as information regarding promoter/non-promoter peak locations.

overlap (*Figure 2B–D*). Dref (DNA replication-related element-binding factor) is a BED finger-type transcription factor shown to bind to the sequence 5'-TATCGATA (*Hirose et al., 1993*), a highly conserved sequence in the core promoters of many *Drosophila* genes (*Ohler et al., 2002*), whereas Cad encodes a homeobox transcription factor that is maternally provided and forms a concentration-gradient enriched at the posterior (*Mlodzik et al., 1985*). Cad exhibits preferential activation of DPE-containing promoters (*Shir-Shapira et al., 2015*). Trl/GAF is a transcriptional factor that regulates chromatin structure by promoting the open chromatin conformation in promoter gene regions, with optimal binding to the pentamer 5'-GAGAG-3' (*Chopra et al., 2008*).

Called peaks for the Opa 3 hr and Zld nc14 late samples were also compared using HOMER, and analysis revealed similar trends with the Zld nc13-14 earlier sample (*Figure 2—figure supplement 1B*). The main difference was that Caudal is no longer identified as the second-most significant site in the Zld-only peak class associated with Zld nc14 late. Collectively, these results support the view that distinct sets of transcription factors serve to facilitate the different functions of Opa and Zld over time in the embryo (see Discussion).

Furthermore, a direct comparison of the Opa (3 hr) and Zld ChIP-seq occupancy at nc13-14, through aggregation plots, suggests that these two transcription factors can bind to the same enhancers (e.g. *Figure 2G*) as well as independently, to distinct enhancers (e.g. *Figure 2E,F*). Indeed, the respective sites appear explanatory for the observed in vivo occupancy to DNA sequences as the matches to the consensus sequences correlate with the center of the peak (*Figure 2E–G*). The widespread binding of Opa in the genome supports the view that this factor functions broadly to support gene expression, as previously suggested from ChIP-chip studies for a number of other transcription factors in the early embryo (X.-Y. *Li et al., 2008*). Therefore, we undertook an analysis of gene expression changes associated with knockdown of *opa*, in particular to assess whether it generally impacts patterning.

## RNA-seq from *opa* RNAi embryos shows that gene expression is regulated by Opa along both axes at cellularization

To generate homogenous populations of mutant embryos, we used RNAi to knockdown levels of *opa*. Embryos were depleted of *opa* transcript by expression of a short hairpin (sh) RNAi construct at high levels using MTD-GAL4, a ubiquitous, maternal driver that is also active in early embryos (*Figure 3A*; *Petrella et al., 2007*; *Staller et al., 2013*). This same approach was used previously to perform *zld* sh RNAi (*sh_zld*) (*Sun et al., 2015*).

Using an anti-Opa antibody, we confirmed that protein levels are greatly diminished upon *opa* short hairpin (sh) RNAi (*sh_opa*) but are retained, though slightly reduced in *sh_zld* (*Figure 3A*; ~1.4 fold reduction, see Materials and methods). This result suggests that *opa* expression is only partially under Zld regulation, and indicates these factors may have separable roles. We also compared gene expression between *opa1* mutants (*Benedyk et al., 1994*; *Cimbora and Sakonju, 1995*) and *sh_opa* embryos by performing in situ hybridization to visualize transcripts of representative Opa target genes *sog* and *engrailed* (*en*) (see *Figure 1* and *Clark and Akam, 2016*; *Benedyk et al., 1994*). We found that expression phenotypes for *sog* and *en* were similar in these two *opa* mutant genotypes (*Figure 3B*).

In order to assay Opa's broad effects on gene expression, RNA-sequencing (RNA-seq) analysis was performed on single-embryo control (*yw*) and knockdown (MTD-Gal4, *sh_opa*) samples, carefully staged to nc14D (see Materials and methods). Up- and down-regulated genes were identified and the data visualized as a heatmap with Z-score representing relative expression value across replicates (*Figure 3C*). Specifically, RNA-seq data support the view that Opa regulates zygotic expression of genes broadly in embryos at nc14 (*Figure 3D*; *Figure 3—figure supplement 1F*); over 667 genes were found to be significantly downregulated (adjusted p value<0.05; *Figure 3—source data 1*) and 36.8% are Opa-only bound targets (*Figure 3—source data 2*). Surprisingly, despite Opa's canonical role as an activator, we also found that 350 genes were significantly upregulated upon *opa* knockdown (adjusted p value<0.05; *Figure 3—source data 1*). Using the DVEX database of spatial transcript expression patterns inferred from single-cell sequencing of the stage six embryo (just following the late nc14D timepoint analyzed by our RNA-seq analysis) (*Karaiskos et al., 2017*), we found that affected genes in *sh_opa* embryos exhibit a wide variety of patterns that span both the DV and AP axes, including terminal regions, in both up- and down-regulated classes (*Figure 3E*). In particular, the genes *doc2* and *doc3* expressed in the presumptive dorsal ectoderm at this stage

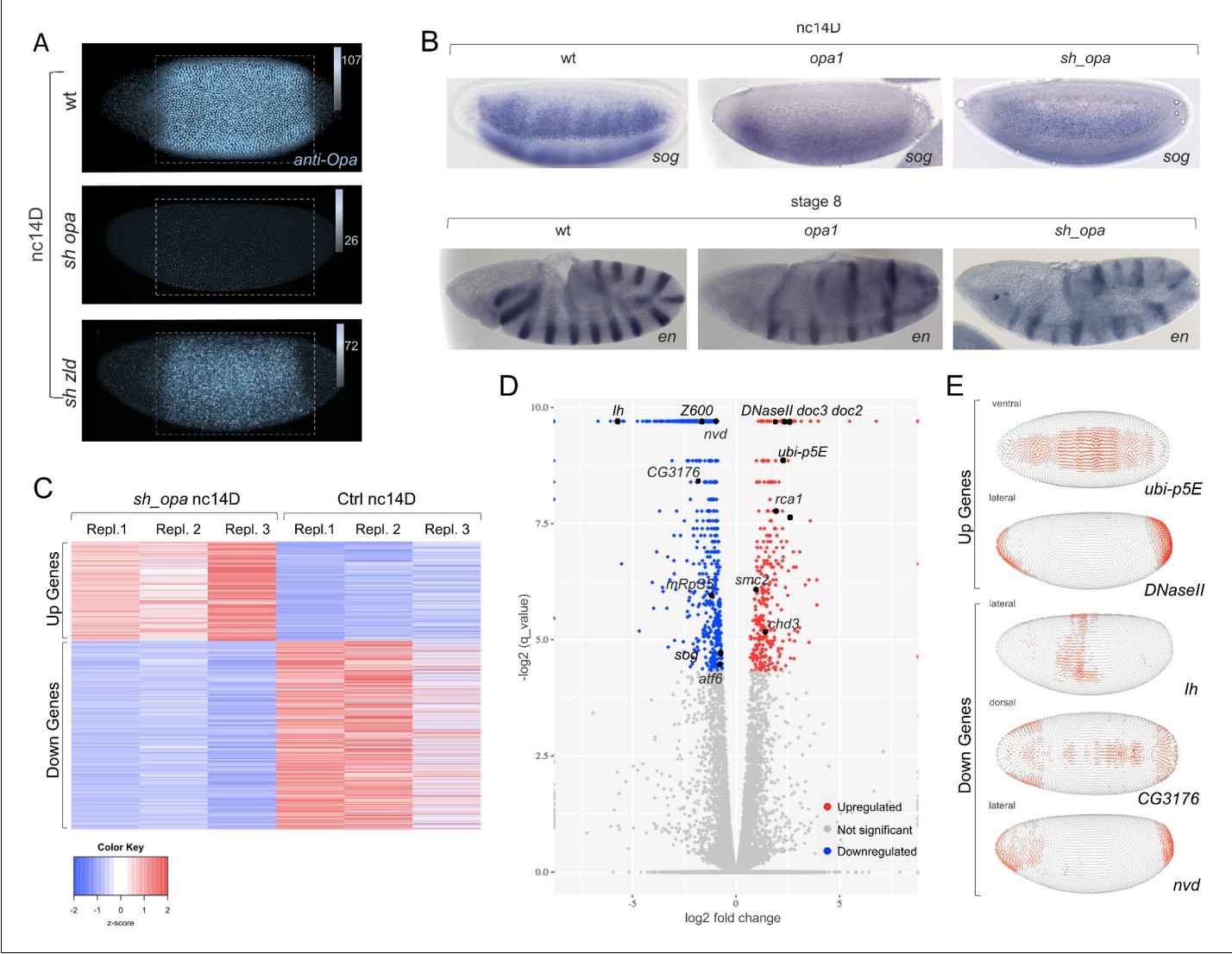

**Figure 3.** *opa* mutants broadly affect gene expression in nc14, preceding segmentation, suggesting a more general role for this gene. (A) Anti-Opa antibody staining (cyan) of wildtype (wt; **A**), *opa* RNAi (*sh_opa*), or *zld* RNAi (*sh_zld*) embryos (n = 3–5 per genotype) at nc14D. The selected area (grey rectangular box) was quantified using ImageJ/Fiji (see Materials and methods). Gradient bars and numbers in upper right corners represent the intensity of each fluorescent image's selected area. (B) In situ hybridization using riboprobes to *sog* at nc14D, as well as *en* staining at stage eight in wt, *opa1* mutant and *sh_opa*.MTD-Gal4 embryos. (C) RNA-seq analysis was performed using control (*yw* females crossed to *sh_opa* males) and *sh_opa* embryos at nc14D (n = 3 per genotype). Replicate expression of up- and down-regulated genes is presented as a heatmap with Z-score representing relative expression value across replicates. Color-key: blue represents low expression and red high expression. This plot demonstrates consistency of RNA-seq results across different replicates. (D) Volcano plots for genes identified through RNA-seq to be significantly downregulated (left; blue versus grey) or significantly upregulated (right; red versus grey) genes. Subset of genes that exhibit Zld and/or Opa occupancy are noted; see also *Figure 3— source data 1*. (E) Images from the DVEX virtual expression software show expression patterns of some of the differentially expressed Opa/Zld or Opa-only targets (*Karaiskos et al., 2017*).

The online version of this article includes the following source data and figure supplement(s) for figure 3:

**Source data 1.** Single embryo RNA-seq data associated with *opa* RNAi versus control nc14D embryos.
**Source data 2.** Association of genes identified by *sh_opa* RNA-seq with Zld and/or Opa ChIP-seq peaks.
**Figure supplement 1.** Opa is responsible for differential gene expression in both embryonic axes as early as nc14.

(*Reim et al., 2003*) are significantly upregulated in *sh_opa* embryos (*Figure 3D* and *Figure 3— source data 2*). We noticed that *opa* mutants also exhibit a u-shaped/tailup cuticular phenotype (weak relative to strong pair-rule phenotype; *Jürgens et al., 1984*; *Benedyk et al., 1994*), typically

relating to problems with dorsal ectoderm patterning and, later, germ band retraction (e. g. *Reim et al., 2003*).

To determine if observed changes in expression relate to direct action of Opa, we assessed whether Opa binding was associated with affected genes. Opa (3 hr) ChIP-seq peaks were identified in promoter-proximal regions [transcription start site (TSS) ±3 kb] of both up- and down-regulated genes (*Figure 3—figure supplement 1A*). Some of these regions were also co-occupied by Zld later at cellularization, but more so for the upregulated gene set (*Figure 3—figure supplement 1B, C, F*). Gene Ontology (GO) analysis for these two sets of genes also demonstrate that upregulated genes tend to relate to nervous system development/neurogenesis; whereas downregulated genes tend to relate to cellular processes such as biogenesis of organelles and metabolism (*Figure 3—figure supplement 1D, E*). As Zld has been shown to promote neurogenesis (*Liang et al., 2008*), it is possible that Opa and Zld have antagonistic roles that coordinate the MZT (see Discussion).

## Opa ChIP-seq peaks are associated with late-acting enhancers driving expression along both axes

Within the total set of 16085 Opa ChIP peaks observed at the 3 hr timepoint we found, surprisingly, that Opa is associated with genes expressed along both the anterior-posterior (AP; *Figure 4A–B', D–E'*) and dorsal-ventral (DV; *Figure 4C* and *Figure 4—figure supplement 1A*) axes. These targets include, but are not limited to, genes involved in segmentation (e.g. *oc* and *slp1*: *Figure 4—figure supplement 1B, C*) as predicted by previous studies (e.g. *Clark and Akam, 2016*; *Prazak et al., 2010*).

We also found evidence in occupancy trends for Opa and Zld that suggest both these factors influence the timing of enhancer action. Opa binding at the 3 hr timepoint is associated with enhancers that are initiated in nc14: *hb_stripe* (*Figure 4A'*; *Koromila and Stathopoulos, 2017*; *Perry et al., 2012*), *slp1_DESE* (*Figure 4—figure supplement 1C*; *Prazak et al., 2010*), and *oc_Prox* (*Figure 4—figure supplement 1B*; *Chen et al., 2012*). In contrast, Zld binding during the same stage, encompassed by Zld nc13-14 and nc14 late ChIP-seq, is associated primarily with enhancers active earlier such as the *hb_stripe* enhancer, whereas the *hb_HG4-7* enhancer active later is not bound by Zld though it is associated with Opa (*Figure 4A*). Similarly at the *even skipped* (*eve*) and *rhomboid* (*rho*) loci, late-acting enhancers (i.e. *eve_LE* and *rho_SHA*) are bound predominantly by Opa; whereas, enhancers active earlier (i.e. *eve3/7*, *rho_NEE* or *oc_Distal*) receive input from Opa and Zld or Zld only (*Figure 4B,B'* and *Figure 4—figure supplement 1B, D*). While Opa is associated with late-acting enhancers, we can not dismiss a role for Zld at later stages, as, for example, the VT40842 enhancer associated with the gene *single-minded* (*sim*) is active later (stage 6 onwards) and is bound by both Opa and Zld at an earlier stage (*Figure 4C* and *Figure 4—figure supplement 1A*).

Furthermore, we found evidence that Opa is preferentially associated with promoters, whether or not Zld is co-associated (*Figure 4—figure supplement 2A*) as demonstrated by calculating the distances of ChIP-seq peak centers to TSS for both the Opa (3 hr) and (4 hr) samples (*Figure 4—figure supplement 2B*). While Zld was shown to preferentially associate with promoters at a much earlier stage (nc8; *Harrison et al., 2011*), we found that binding of Zld to Zld-only enhancers occurs in more distal regions at stage 5 (i.e. nc13/14 and nc14 late samples) (*Figure 4—figure supplement 2A*). It is possible that once Opa is expressed it preferentially associates with promoter regions and either competes and/or co-regulates with Zld (see Discussion).

## H3K4me3 and H3K4me1 histone marks are enriched at nc14 at regions occupied by Opa

Previous genomics studies have demonstrated that particular histone marks correlate with active enhancers at different developmental stages in the early embryo. For example, there is a dramatic increase in the abundance of histone modifications at the MZT, coinciding with zygotic genome activation (*Schulz and Harrison, 2019*). We investigated whether Opa-only, Zld-only, and Opa-Zld overlap regions exhibit differences in chromatin marks that might support our hypothesis that Opa-associated regions are active later than Zld-only regions. For the purposes of this analysis, 10 published ChIP-seq datasets relating to histones or histone modifications (X.-Y. *Li et al., 2014*) were assayed for coincidence of any marks with Opa- and/or Zld-bound regions identified by our analysis (see Materials and methods). Only H3K4me3 and H3K4me1 histone marks were found to differ

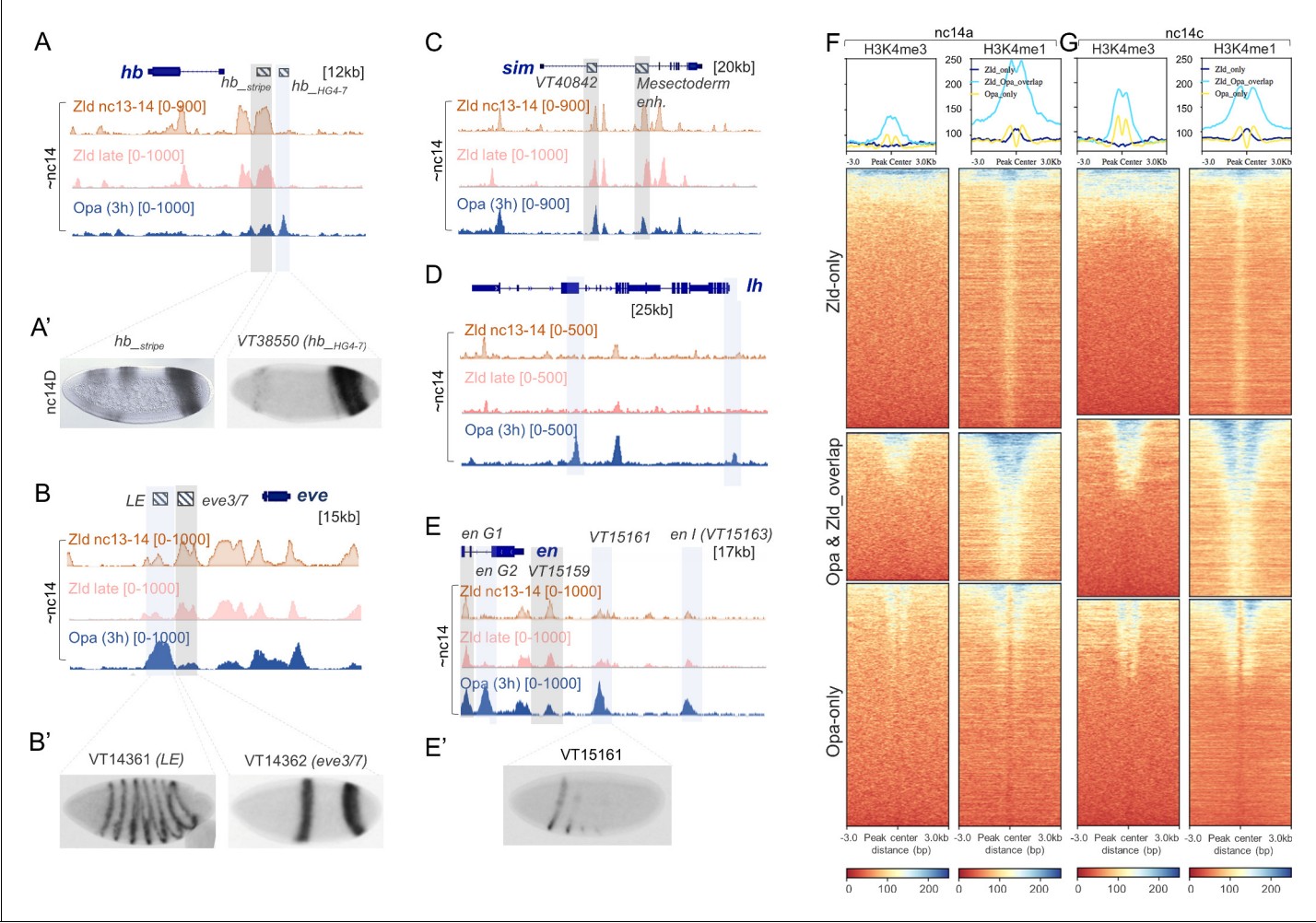

**Figure 4.** Opa chromatin immunoprecipitation (ChIP) demonstrates binding globally including enhancers active at nc14 as well as later stages. (A–E')
In house (A', *hb_stripe* ; *Perry et al., 2012*; *Koromila and Stathopoulos, 2017*) and publicly available genome-scale enhancer characterization (VT
reporters; *Kvon et al., 2014*) demonstrating expression at nc14D by in situ (A', B', E'), as well as IGV browser tracks of genes expressed along either
the AP (A, B, D, E) or DV (C) axes showing Zld nc13-14 (orange), Zld nc14 late (pink), and Opa (3 hr) (blue) ChIP-seq replicates (as indicated). Anti-Opa
antibody was used to immunoprecipitate chromatin isolated from embryos ~ 3 hr in age (see Materials and methods). Published Zld ChIP-seq data for
two different timepoints is shown (GSM763061: nc13-14 and GSM763061:nc14 late; *Harrison et al., 2011*). Nc14 was used as a point of comparison
between the 3 ChIP samples. Gray boxes indicate regions with significant occupancy by both Opa and Zld as detected by ChIP-seq peaks, which can
be located at promoter and/or distal regions (A, C, D, F); whereas, light blue boxes indicate regions with significant Opa-only binding at promoter and/
or distal regions (A, B, E). (F, G) Heatmaps produced by deepTools (see Materials and methods) were used to plot histone H3K4me3 and H3K4me1 at
nc14a (G) and H3K4me3 and H3K4me1 at nc14C (H) signal intensities centered at different ChIP-seq regions (Zld-only, Opa-Zld overlap, and Opa-only
bound ChIP-seq peaks). For the two different timepoints nc14A and nc14C, different Zld ChIP data were used (GSM763061: nc13-nc14 and nc14 late,
respectively; *Harrison et al., 2011*). Key indicates histone signal intensities (deepTools normalized RPKM with bin size 10). For this and all subsequent
data presented using heatmaps, the first sample in the heatmap was used for sorting the genomic regions based on descending order of mean signal
value per region; all other comparison samples were plotted using the same order determined by the first sample.

The online version of this article includes the following figure supplement(s) for figure 4:

**Figure supplement 1.** Additional examples of binding of Opa and/or Zld to enhancer regions active in nc14 and later.

**Figure supplement 2.** Position of ChIP-seq called peaks relative to transcription start sites (TSSs) as well as a comparison of Opa-early (3 hr) versus
Opa-late (4 hr) peak size and overlap.

between Opa- versus Zld-bound peaks (*Figure 4F,G*). Both histone marks are first detectable at the
MBT, while absent prior to nc14a, whereas their associated genes are considered to be activated at
later stages (X.-Y. *Li et al., 2014*; *Chen et al., 2013*).

Heatmap modules of deepTools (see Materials and methods) were used to calculate and plot histone H3K4me3 and H3K4me1 signal intensities assayed at two timepoints, nc14A and nc14C, for different ChIP-seq peak sets: Opa-only; Zld-only; or Opa-Zld overlap. Our analysis shows that Zld-only bound regions are depleted for H3K4me3, as shown previously (X.-Y. *Li et al., 2014*), as well as for H3K4me1 at both time points relative to Opa-only or Opa-Zld overlap bound regions (*Figure 4F, G*). The higher levels of H3K4me1 in the Opa-bound peaks could reflect a poised state of late-acting enhancers, relate to spatial regulation (e.g. repression), and/or support enrichment of Opa-binding at promoters (*Figure 4—figure supplement 2A, D*; *Bonn et al., 2012*; *Koenecke et al., 2017*; *Rada-Iglesias et al., 2011*).

## *opa* knockdown results in global changes in chromatin accessibility

We hypothesized that Opa functions as a pioneer factor to regulate temporal gene expression starting at nc14 in the celluarizing blastoderm. To test this, we investigated whether Opa functions to regulate chromatin accessibility genome-wide. We used ATAC-seq (Assay for Transposase-Accessible Chromatin using sequencing; *Buenrostro et al., 2015*) to investigate the state of chromatin accessibility in *opa* RNAi and *opa1* zygotic mutants compared to wild type by assaying carefully-staged individual embryos (see Methods).

To provide insight into the potentially different roles of these two transcriptional factors, Opa and Zld, ATAC-seq analysis was conducted on single *sh_opa*, *opa1* mutant, *sh_zld,* or 'wt' (control: *opa* sh without Gal4 driver) embryos and the results compared (see Materials and methods for details). To start, we determined the relative accessibility indices of embryos in wt versus *sh_opa* for Opa (3 hr) ChIP-seq peak regions as well as subclasses: Opa-only, Zld-only, or Opa-Zld overlap regions (i.e. *Figure 2A*) using deepTools (*Ramírez et al., 2014*) with RPKM method for normalization (see Materials and methods). A general decrease in chromatin accessibility was associated with the *sh_opa* sample relative to wt in nc14D embryos (*Figure 5B* and *Figure 5—figure supplement 1D*). Of the Opa-bound ChIP-seq defined peak regions, those also occupied by Zld (i.e. Opa-Zld overlap regions) had, on average, ~2 fold higher ATAC-seq signal (i.e. accessibility) in wt than those bound solely by either Zld or Opa (i.e. Zld-only or Opa-only) (*Figure 5—figure supplement 1D*). *opa* RNAi (*sh_opa*) decreases accessibility at Opa-only regions but also at the Opa-Zld overlap bound regions (*Figure 5—figure supplement 2*). In summary, occupancy of both Opa and Zld is a better indicator of open chromatin regions than either factor alone; and, surprisingly, Opa regulates chromatin accessibility at Opa-only as well as Zld-co-bound regions. As Zld has been documented to function as a pioneer factor that helps to make chromain accessible, these results suggest that Opa may also function in this role.

We analyzed chromatin accessibility at an earlier timepoint, nc14B, finding that in wt the Opa-bound regions are less accessible at nc14B compared to nc14D, and investigated whether Opa levels relate to the timing of this change in accessibility (*Figure 5A*, left, compare with *Figure 5B*). To do this, we ectopically-expressed full-length *opa* again using the maternal-zygotic MTD-Gal4 driver to ensure strong early embryonic expression. Single embryos of stage nc14B that ectopically express *opa* (UAS-*opa*) in this fashion exhibit a clear increase in chromatin accessibility across Opa-bound regions (*Figure 5A*) suggesting that Opa acts to open chromatin.

We therefore hypothesized that Opa functions as a pioneer-like factor supporting previously unattributed Zld-independent facilitation of zygotic genome activation. In studies of Zld accessibility using an alternate method, FAIRE-seq, it was determined that, while Zld is clearly important for facilitating chromatin accessibility in the early embryo, it is not the only factor that supports this function; some chromatin regions remain accessible in *zld* mutants (*Schulz et al., 2015*). Opa is expressed in *sh zld* mutants, although at reduced levels (*Figure 3A*). However, all Opa-bound regions - both Opa/Zld overlap and Opa only regions - are not affected upon *zld* knock-down (*sh_zld*) (*Figure 5—figure supplement 3A-C, E-G*), whereas Zld only regions are decreased in accessibility but only at an earlier timepoint, nc13, not at nc14 (*Figure 5—figure supplement 3H* compare with 3D). While our results demonstrate that Opa, not Zld, regulates chromatin accessibility at nc14, we cannot exclude an accessory role for Zld.

To investigate Opa's mechanism of action, we compared the distance of *opa* motif to the nearest nucleosomes at nc14B between wildtype embryos and those that ectopically express *opa* to determine if evidence of nucleosome displacement could be inferred. ATAC-seq fragment sizes reflect nucleosome organization with a peak in the fragment-size distribution at 120–200 bp arising from

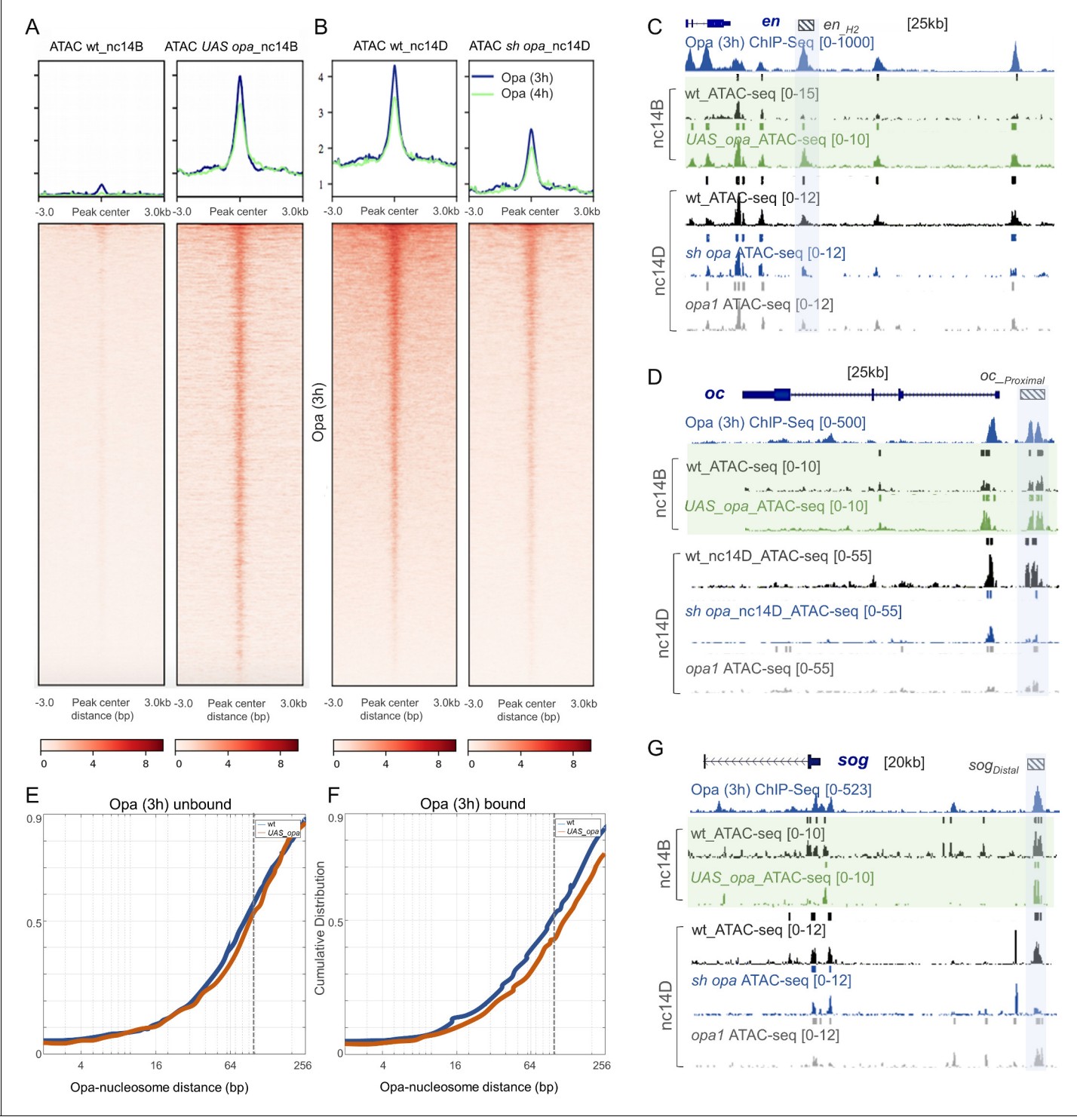

**Figure 5.** Opa influences chromatin accessibility during nc14 in *Drosophila* embryos in part by displacing nucleosomes. (**A, B**) Heatmaps of normalized paired-end ATAC-seq signal from nc14D wt, *sh_opa* and *opa1* mutant embryos for the regions called from Opa (3 hr) ChIP-seq. Each row of the heatmap is a genomic region, centered to peaks of accessibility signals. The accessibility is summarized with a color code key representative of no accessibility (white) to maximum accessibility (red). Plot at the top of the heatmap shows the mean signal at genomic regions centered at peaks of accessibility signals (Opa 3 hr: blue trace; Opa 4 hr: green trace). Averaging of ATAC-seq data from two nc14D embryos (n = 3) were used for this analysis (see Materials and methods). (**C, D, G**) UCSC dm6 genome browser tracks of representative loci showing Opa (3 hr) (navy blue), ChIP-seq replicates, as well as single replicates of nc14B (green box) and nc14D ATAC-seq. Examples of late enhancer regions that significantly gain/lose accessibility, compared to wt, in either *UAS-opa*, *sh_opa* and/or *opa1* mutants are defined by blue shaded regions. Plots show mean normalized read

*Figure 5 continued on next page*

*Figure 5 continued*

coverage of the replicates (see also *Figure 5—source data 1*). (E, F) Cumulative distribution of measured distances between 21440 unbound (E) and 4481 bound (F) Opa motif sites and modeled nucleosome dyad positions (*Schep et al., 2015*) under wildtype conditions (blue) or upon ectopic expression of *opa* (red). The expected coverage of a nucleosome is depicted by the vertical dotted line. X-axis is log2 scaled.

The online version of this article includes the following source data and figure supplement(s) for figure 5:

**Source data 1.** Single embryo ATAC-seq data for multiple genetic backgrounds and stages (see tabs) to investigate relationship of Opa to chromatin accessibility.

**Figure supplement 1.** Opa or Opa+Zld are required for chromatin accessibility as revealed by ATAC-seq on single embryos.

**Figure supplement 2.** Chromatin accessibility changes associated with Opa ChIP-detected binding as related to Opa/Zld occupancy.

**Figure supplement 3.** Single-end ATAC-seq confirms Zld's role early (nc13) but with minimal effect later (nc14).

DNA protected by a nucleosome. Ectopic expression of *opa* results in a trend toward shift in the positions of nucleosomes relative to binding sites (*Figure 5—figure supplement 1G, H*), suggesting that Opa displaces nucleosomes. To obtain more definitive evidence, we performed a quantitative analysis of our ATAC-seq data based on modeled nucleosome positions. The cumulative distribution of measured distances between 21440 unbound (*Figure 5E*) and 4481 bound (*Figure 5F*) Opa motif sites and modeled nucleosome dyad positions was determined using previously defined methods (see Materials and methods; *Schep et al., 2015*). We observe a shift to larger motif-nucleosome distance upon ectopic expression of *opa* compared to wildtype (*Figure 5E,F*; red versus blue lines, respectively). These data support the view that Opa occupancy on DNA displaces nucleosomes. A recent study found that alternatively in *opa* mutant embryos there is a shift to smaller distance, which also supports the view that Opa is required to displace nucleosomes (*Soluri et al., 2020*).

## Opa-only occupied peaks require Opa to support accessibility at mid-nc14

Chromatin accessibility as characterized by single-embryo ATAC-seq revealed 88.5% overlap between the *opa1* and *sh_opa* closed chromatin peaks (versus open peaks in control), as well as 75% overlap between the *sh_opa* closed (accessible/open peaks in control) and *UAS-opa* open peaks (non-accessible/closed peaks in control) (*Figure 5—figure supplement 1A*). Accessibility was also examined at particular enhancers known to be bound by Opa. In particular, enhancer regions active in nc14 for genes expressed along AP and DV axes exhibit Opa-dependent changes in accessibility. For example, the *VT15161 en* enhancer (*Kvon et al., 2014*) exhibits an increase in accessibility in response to higher Opa levels, but a decrease in both *sh_opa* and *opa1* mutants (*Figure 5C*, blue shaded region). Similar trends were identified for *oc_Proximal, sog_Distal,* and *hb_stripe* enhancers (*Figure 5D,G* and *Figure 5—figure supplement 1E*; blue shaded regions) (*Perry et al., 2011*; *Koromila and Stathopoulos, 2017*). Moreover, the accessibility of these same enhancers was not affected in *sh_zld* (e.g. *Figure 5—figure supplement 1B, E, F*). On the other hand, accessibility at other enhancer sequences, such as *eve_3–7*, was affected by changes in both *opa* and *zld* (*Figure 5—figure supplement 1C*; grey-shaded box). The Opa-bound regions fall into three classes: (i) regions that require Opa for accessibility (blue-shaded regions); (ii) regions that require both Opa and Zld for accessibility (grey-shaded regions); and (iii) regions that require Zld, but not Opa, for accessibility [mint-shaded regions; e.g. enhancer *sog_intronic* (*Markstein et al., 2002*) and *eve_LE* (*Fujioka et al., 2013*; *Figure 5—figure supplement 1B, C*)]. Collectively, these results support the view that Opa can influence chromatin accessibility, but not all regions that are bound by Opa require this factor for accessibility (i.e. class iii). It is possible that at Opa-Zld overlap regions, in which both factors are bound, either Opa or Zld can suffice to support accessibility.

To determine whether the global effects on chromatin accessibility observed in *opa* mutants have consequences for patterning, we examined gene expression in mutant embryos and assayed for patterning phenotypes. We performed in situ hybridizations on wildtype and *sh_opa* embryos using riboprobes to detect endogenous transcripts for the genes *sog* and *sna*, expressed along the DV axis, and for *hb*, expressed along the AP axis. *zld* RNAi (*sh_zld*) mutants were also examined for comparison; loss of *zld* is known to affect both *sog* and *hb* as well as to cause a delay in *sna* expression that recovers by nc14 (*Nien et al., 2011*; *Liang et al., 2008*). In addition, embryos were examined at two stages, nc13 and nc14B, corresponding to timepoints before Opa is expressed or when it first initiates, respectively (e.g. *Figure 1D*). Even early, at nc13, *zld* mutant embryos exhibit loss of

expression for all three genes examined (*hb, sna,* and *sog*), supporting the view that Zld is necessary for early gene expression (*Figure 6C,E*). In contrast, little difference in expression was observed in *opa* mutants at nc13. This is unsurprising as *opa* is not expressed at nc13 and therefore would not be expected to affect patterning at this stage (*Figure 6C*). Later, at nc14C, *opa* mutants do exhibit expression defects, as both *sog* and *hb* expression is diminished (*Figure 6D*). These expression defects likely relate to lack of Opa input at *sog_Distal*, *hb_stripe*, and *hb_HG4-7* enhancers that exhibit Opa-dependent changes in accessibility (*Figure 5G*; *Figure 4A,A′*).

In addition to these findings relating to patterning, we also observe temporal bias in Opa's genomic effects. ATAC-seq data for nc14B individual embryos in which *opa* was ectopically expressed (UAS-*opa*) exhibit a significant increase in chromatin accessibility at regions bound by Opa early (ChIP-seq 3 hr; *Figure 6—figure supplement 1A-C*). However, ectopic expression of *opa* failed to increase accessibility at late-only regions bound by Opa (ChIP-seq 4 hr peaks not also present early) when assayed at nc14B (*Figure 6—figure supplement 1A, D*). Furthermore, only ChIP-seq regions that were present at the early timepoint exhibited decreased accessibility upon *opa* RNAi (i.e. 3 hr peaks as well as 3 hr + 4 hr overlap peaks). These data further support the view that Opa binding is dynamic and associated with changes in accessibility.

Finally, to provide additional insight into Opa's function, we examined how Opa-dependent changes in chromatin accessibility correlate with changes in gene expression. Opa ATAC-seq peak regions were associated with the nearest gene transcription start site (TSS) to calculate overlap gene counts with *opa* mutant RNA-seq up- and down-regulated genes for nc14D embryos. Surprisingly, we found that Opa-dependent changes in chromatin accessibility occur near genes that are downregulated as well as those that are upregulated upon *sh_opa* (*Figure 6A,B*; see Discussion).

## Discussion

In summary, our experiments indicate that Opa is a late-acting timing factor, and likely pioneer factor, which regulates gene expression throughout the embryo, along DV as well as AP axes. Opa, a non-canonical broadly expressed pair-rule gene, has only previously been implicated in AP axis patterning, but our initial analysis of the *sog_Distal* enhancer, which is active along the DV axis, led us to investigate a more general role for this factor. We show that *opa* mutants exhibit broad DV patterning changes in addition to previously identified AP patterning phenotypes and its role as a broadly-acting regulator of gene expression is supported by whole-genome gene expression profiling of *sh_opa* mutant embryos by RNA-seq. Opa likely acts directly to regulate gene expression, as Opa chromatin immunoprecipitation (ChIP) demonstrates widespread binding throughout the genome, including at the *sog_Distal* enhancer. Additional data from single-embryo ATAC-seq provide insight into the mechanism by which Opa supports gene expression as *opa* knockdown affects chromatin accessibility at regions occupied by Opa but not by Zld, as determined by ChIP analysis. Chromatin accessibility in early embryos appears to be predominantly supported by Zld and then, once Opa is expressed in mid-nc14, the two factors seem to work together in this role. However, zygotic genes (or particular enhancers) associated with mid-nc14 to gastrulation or later, appear to be preferentially bound by Opa. Therefore, we suggest that Opa acts following the pioneer factor Zld to influence timing of zygotic gene activation at the whole-genome level by actively increasing late-acting enhancer accessibility (*Figure 6E*). Furthermore, Opa-bound regions (with and without Zelda) are enriched for late histone methylation, and this may relate either to Opa's preference promoters or suggest a more causative function for Opa with respect to chromatin state. Opa's regulatory impacts during development may therefore include both control of epigenetic marks as well as chromatin accessibility at promoters and/or promoter-proximal cis-regulatory elements. Further, in addition to supporting gene expression in a general manner by making chromatin accessible, Opa also presumably influences the activity of other transcription factors such as Run when co-bound to enhancers through mechanisms that are as yet not completely understood (*Koromila and Stathopoulos, 2019*; *Prazak et al., 2010*).

Through analysis of overrepresented motifs in the Opa ChIP-seq peak regions using the program HOMER, we identified a number of factors that may co-associate with Opa and provide additional insight into its function. For example, a motif matching the Dref binding consensus is enriched at Opa ChIP-seq peaks. Dref is highly enriched at promoters and has been implicated in multiple roles during *Drosophila* development including regulation of expression of signaling pathway components

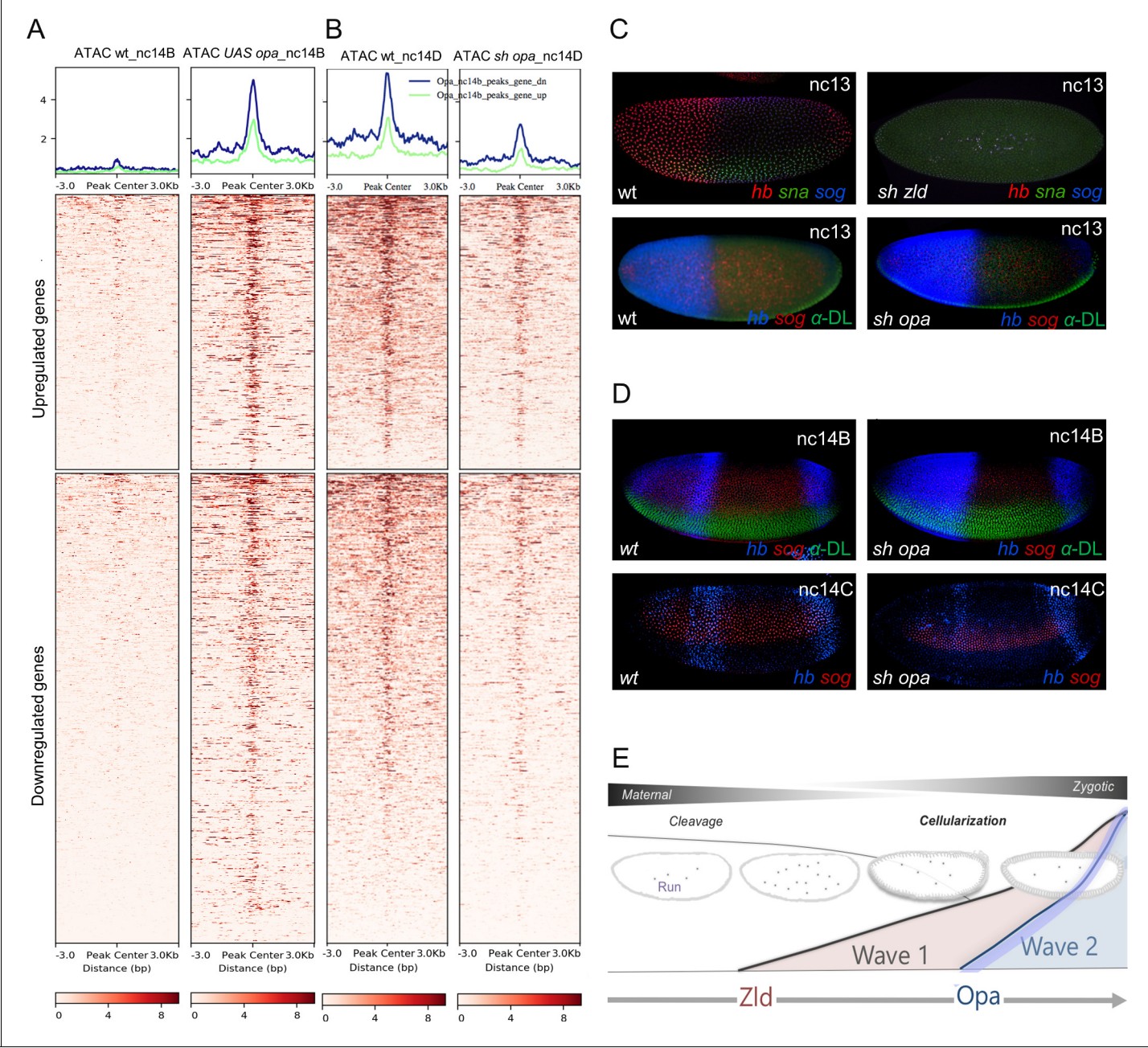

**Figure 6.** Opa is a late-acting, pioneer factor whose action follows Zelda to drive a second wave of zygotic gene expression. (A, B) Aggregated signals and heatmaps of nc14B normalized ATAC-seq signal from wt and *UAS-opa*, as well as nc14D wt and *opa* RNAi (*sh_opa*) mutant embryos for downregulated (blue trace) and upregulated target genes as identified by RNA-seq (green trace). Each row of the heatmap is a genomic region, centered to peaks of accessibility signals. The accessibility is summarized with a color code key representative of no accessibility (white) to maximum accessibility (red). Plot at the top of the heatmap shows the mean signal at genomic regions centered to peaks of accessibility signals (A). (C, D) In situ hybridization using riboprobes to *hb*, *sna*, and/or *sog*, as well as anti-Dorsal staining (where noted to highlight ventral regions) of wt and *sh_opa* embryos at indicated stages (n = 5 per genotype). (E) Schematic illustrating a model supported by our results, which is that Opa, a general timing factor and likely a late-acting pioneer factor, drives a secondary wave of zygotic gene expression, following and coordinating with Zelda, to support the maternal-to-zygotic transition.

The online version of this article includes the following figure supplement(s) for figure 6:

**Figure supplement 1.** Opa changes in chromatin accessibility in *opa* overexpression and *opa* shRNAi embryos in regions identified by Opa (3 hr) and/ or (4 hr) ChIP-seq time points.

and chromatin organization including insulator and chromatin remodeling functions (rev. in *Tue et al., 2017*). In particular, as Dref has been linked to insulator function (*Gurudatta et al., 2013*) and Zld has been shown to be associated with locus-specific TAD boundary insulation (*Hug et al., 2017*), future studies should examine whether temporal progression of gene regulatory networks is supported not only by the sequential action of pioneer factors to affect chromatin accessibility at enhancer/promoters but also by influencing chromatin conformation.

Regions of chromatin accessibility are thought to be established sequentially, where enhancers are opened in advance of promoters and insulators (*Blythe and Wieschaus, 2016a*) and our results show enrichment of Opa binding near promoters. Opa may therefore influence gene expression timing by affecting accessibility at promoters, possibly, in combination with Dref. Alternatively, while the Trl/GAF motif was associated with all classes of peaks, it was enriched in Zld-only peaks. Furthermore, Trl/GAF and Dref were shown to be associated with early versus late embryonic expression (stage five and later), respectively (*Darbo et al., 2013*; *Schulz et al., 2015*; *Hochheimer et al., 2002*). These results collectively support the view that Opa, like Dref, is a late-acting factor, and that Trl/GAF may work early with Zelda. However, other studies have shown that Trl/GAF acts coordinately but separately from Zld to support chromatin accessibility in regions that do not require Zld (*Moshe and Kaplan, 2017*).

More generally, the triggering of temporal waves of gene regulation in response to chromatin accessibility changes is a potentially widespread mechanism used to control developmental progression. A key future pursuit will be to understand how the expression of genes that act as triggers, such as *opa*, are regulated. To start, the *opa* transcript is over 17 kB in length, and this relatively long transcript size may contribute to its expression in late nc14 as previously studies have suggested that transcripts of this length are difficult (or impossible) to transcribe in earlier nuclear cycles due to short interphase length (i.e. nc13, etc) (*Sandler et al., 2018*; *Shermoen and O'Farrell, 1991*). *opa* expression is also regulated by the nuclear-cytoplasmic (N/C) ratio, whereas other genes like *snail* are not sensitive to N/C ratio but appear instead to be regulated by a maternal clock (*Lu et al., 2009*). It has been hypothesized that the initiation of N/C ratio-responsive genes is regulated by a maternal repressor, the activity of which is counteracted by increasing N/C ratio possibly through dilution (e.g. *Hamm and Harrison, 2018*). Therefore, as a N/C-ratio responsive gene, *opa* expression may be regulated by maternal repression.

We propose that genes of different timing classes may be preferentially regulated by different pioneer factors, and that multiple factors may act coordinately to control stage-specific gene expression. For example, at nc14, which encompasses the MBT, Opa appears to be one timing factor acting, but Zld and possibly also other factors support this process. Despite significant expression changes in important regulators of this process such as *Z600/frühstart* (*frs*; *Grosshans et al., 2003*) in *sh_opa* embryos, no gross cellularization defects or changes in length of the cell cycle were detected. However, late *Z600/frs* expression at nc14D is associated with the dorsal ectoderm (see *Figure 2C* in; *Grosshans et al., 2003*), and loss of Opa may present defects at later stages possibly in relation to function of dorsal tissues such as amnioserosa. In addition, Zld regulates expression of *Z600/frs* as well as a number of other genes involved in cellularization such as *slam*, *halo*, *btsz*, *bnk*, *nullo*, *CG14427*, and *Sry-α* (*Nien et al., 2011*; *Lu et al., 2009*). Of these genes, we also found evidence that Opa regulates expression not only of *Z600/frs* but also *slam*, *halo*, and *bnk* (see *Figure 3—source data 1*: RNA-seq data). As *opa* expression is dependent on the ratio of nuclear to cytoplasmic volume (N/C-ratio sensing) (*Lu et al., 2009*), our data support the view that Opa sits at the top of a gene regulatory hierarchy that facilitates levels of expression of the genes above, which are involved in MBT, as well as other N/C-ratio sensing genes including *sog* and *odd-skipped/CG3851* (this study; *Lu et al., 2009*). Contrastingly, degradation of maternal transcripts, also an important part of the MBT, is not N/C-ratio dependent and likely relates to a function of Zld (*Liang et al., 2008*) not Opa. It is possible that Opa supports expression of some but not all genes that regulate MBT. For example, Opa may not be involved in cell cycle pause as associated with *zld* mutants (*Liang et al., 2008*), but may contribute to regulation of other cellular processes. GO analysis of genes downregulated in *sh_opa* mutants suggest that Opa regulates genes involved in biogenesis of cellular components, mitochondria and other organelles, metabolism, and vesicle organization (*Figure 3—figure supplement 1E*).

Furthermore, pioneer factors may exhibit spatially constrained functions. Opa's expression in the embryonic trunk, but exclusion from the termini, suggest that additional late-acting factors may

serve a pioneer role at the embryonic termini. For example, a recent study suggests that the transcription factor Orthodenticle (Otd) functions in a feed-forward relay from Bicoid to support expression of genes at the anterior of embryos (*Datta et al., 2018*). Furthermore, our analysis of the Zld ChIP-seq dataset identified enrichment for Cad transcription factor binding motifs in Zld-bound regions. Cad is localized in a gradient that emanates from the posterior pole (*Mlodzik et al., 1985*). It is possible that these factors, Otd and Cad, function to support the late-activation of genes expressed at the anterior and posterior termini, possibly also functioning as pioneer factors to facilitate action of particular late-acting enhancers in these domains where Opa is not present.

In addition, studies have shown that Zld influences morphogen gradient outputs (*Foo et al., 2014*), and perhaps Opa does as well, during the secondary wave of the MZT. For example, activation of BMP/TGF-β signaling in the early embryo depends on the formation of a morphogen gradient of Decapentaplegic (Dpp) that corresponds in time with Opa expression (rev. in *Stathopoulos and Levine, 2005*; *Sandler and Stathopoulos, 2016*). We suggest that temporally-regulated activators such as the late-acting, timing factor Opa may also support additional nuanced roles in the temporal regulation of morphogen gradient outputs to support patterning. For example, our previous study showed that BMP/TGF-β target genes exhibit different modes of transcriptional activation, with some targets exhibiting a slower response (*Sandler and Stathopoulos, 2016*). *opa* mutant cuticles exhibit a pair-rule phenotype (*Jürgens et al., 1984*), but upon closer observation also exhibit a weak tail-up phenotype supporting a role for this gene in DV patterning, in particular, for support of BMP signaling target genes including *doc2* and *doc3* (e.g. the u-shaped group; *Wieschaus and Nüsslein-Volhard, 2016*; *Reim et al., 2003*).

Opa is conserved, as it shares extended homology of protein sequence and DNA binding specificity with the Zic family of mammalian transcription factors (*Aruga et al., 1996*; *Hursh and Stultz, 2018*). Zic family members are involved in neurogenesis, myogenesis, skeletal patterning, left-right axis formation, and morphogenesis of the brain (*Grinberg and Millen, 2005*). In addition, Zic family members have been shown to be important for the maintenance of pluripotency in embryonic stem (ES) cells (*Lim et al., 2007*). In particular, Zic3 shares significant overlap with the Oct4, Nanog, and Sox2 transcriptional networks and is important in maintaining ES cell pluripotency by preventing differentiation of cells into endodermal lineages. While we have focused on the role of Opa as an activator of gene expression, it is also possible that Opa acts to limit differentiation paths of cells. The presence of both downregulated and upregulated genes upon *opa* RNAi also supports this view.

## Materials and methods

**Key resources table**

| Reagent type (species) or resource | Designation | Source or reference | Identifiers | Additional information |
|---|---|---|---|---|
| Recombinant DNA reagent | *eve2 promoter-MS2.yellow*-attB | *Bothma et al., 2014* | N/A | |
| Recombinant DNA reagent | *sogD_ΔOpa eve2 promoter-MS2.yellow*-attB | This study | TK61_DNA | |
| Recombinant DNA reagent | *sogD_ΔOpa4 eve2 promoter-MS2.yellow*-attB | This study | TK62_DNA | |
| Recombinant DNA reagent | *sog_Distal_ eve2 promoter-MS2.yellow*-attB | *Koromila and Stathopoulos, 2019* | TK54_DNA | |
| Antibody | anti-Opa (Rabbit polyclonal) | *Mendoza-García et al., 2017* | E990 | IF: 1:200 |
| Antibody | anti-Opa (Rabbit polyclonal) | This study | E992 | IF: 1:200 |
| Antibody | anti-Zelda (Rabbit polyclonal) | This study | N/A | |

*Continued on next page*

*Continued*

| Reagent type (species) or resource | Designation | Source or reference | Identifiers | Additional information |
|---|---|---|---|---|
| Genetic reagent (*D. melanogaster*) | ZH-attP-86Fb | Bloomington *Drosophila* Stock Center (BDSC) | BDSC:23648; FLYB:FBti0076525; RRID:BDSC_23648 | |
| Genetic reagent (*D. melanogaster*) | *sog_Distal* | *Koromila and Stathopoulos, 2019* | TK54 | Transgenic insertion into 86Fb attP |
| Genetic reagent (*D. melanogaster*) | *sogD_ΔRun* | *Koromila and Stathopoulos, 2019* | TK56 | Transgenic insertion into 86Fb attP |
| Genetic reagent (*D. melanogaster*) | *sogD_ΔOpa4* | This study | TK62 | Transgenic insertion into 86Fb attP |
| Genetic reagent (*D. melanogaster*) | *sogD_ΔOpa* | This study | TK61 | Transgenic insertion into 86Fb attP |
| Genetic reagent (*D. melanogaster*) | yw;*Nucleoporin-RFP*;*MCP-NoNLS-GFP* | *Lucas et al., 2013* and *Koromila and Stathopoulos, 2019* | TK59 | |
| Genetic reagent (*D. melanogaster*) | *UAS-shRNA-opa* | BDSC | BDSC:34706: FLYB:FBal0175559: RRID:BDSC_34706 | |
| Genetic reagent (*D. melanogaster*) | *MTD-Gal4* | BDSC | BDSC:31777; FLYB:FBtp0001612; RRID:BDSC_31777 | FlyBase symbol: P{GAL4-nos.NGT} |
| Genetic reagent (*D. melanogaster*) | *opa1* | BDSC | BDSC:3312; FLYB:FBst0305629; RRID:BDSC_3312 | |
| Software, algorithm | JASPAR | *Khan et al., 2018* | http://jaspar.binf.ku.dk/cgi-bin/jaspar_db.pl?rm=browse and db = core and tax_group = insects | |
| Software, algorithm | Imaris 9.0 | | N/A | |
| Software, algorithm | Fiji | *Schindelin et al., 2012* | N/A | |
| Software, algorithm | Bowtie2 | *Langmead and Salzberg, 2012* | | |
| Software, algorithm | MACS2 | *Zhang et al., 2008* | | |
| Other | Halocarbon 27 oil | Sigma-Aldrich | MKBJ5699 | |

## Fly stocks and husbandry

The *y w [67c23]* strain was used as wild type, unless otherwise noted. All flies were reared under standard conditions at 23°C, except for RNAi crosses involving MTD-Gal4 and controls that were reared at 26°C.

For the RNA live imaging experiments, we used the following fly stocks: mRFP-Nup (*Lucas et al., 2013*) and Hsp83-MCP-GFP (*Garcia et al., 2013*). Females homozygous for mRFP-Nup; Hsp83-MCP-GFP were crossed to males containing either the wildtype *sog_Distal* MS2 reporter (*Koromila and Stathopoulos, 2019*) or mutant (i.e. *sogD_ΔOpa* and *sogD_ΔO*pa4, this study; or *sogD_ΔRun*, *Koromila and Stathopoulos, 2019*) MS2 reporters, and imaged live (see below).

*opa1/TM3,Sb* [Bloomington *Drosophila* Stock Center(BDGP)#3312] mutant flies were rebalanced with TM3 *ftz-lacZ* marked balancer. Additionally, embryos were depleted of maternal and zygotic *opa* and *zld* by maternal expression of short hairpin (*sh*) RNAi constructs (*Staller et al., 2013*; *Sun et al., 2015*): UAS-*sh_opa* (passenger strand sequence CAGCTTAAGTACGCAGAATAA targeting *opa* in VALIUM20; TRiP.HMS01185_attP2/TM3, Sb[1]; BDSC#34706) with zero predicted off-targets (*Hu et al., 2013*; *Perkins et al., 2015*), UAS-*sh zld* (passenger strand sequence CGGATGCAAG TTGCAGTGCAA targeting *zld* in VALIUM22) used previously (*Sun et al., 2015*). For RNAi, *UAS-shRNA* females were crossed to *MTD-Gal4* males (BDSC#31777) (*Petrella et al., 2007*; *Staller et al.,*

*2013*). F1 *MTD-Gal4/UAS-shRNA* females were crossed back to *shRNA* males, and F2 embryos collected at 26°C and assayed. For *opa* ectopic expression, *UAS-opa* (*Lee et al., 2007*) females were similarly crossed to *MTD-Gal4* males; F1 *MTD-Gal4/UAS-opa* females were crossed back to *UAS-opa* males, and F2 embryos collected at 26°C and assayed.

In all experiments, both male and female embryos were examined; sex was not determined but assumed to be equally distributed.

## Cloning

The *sog_Distal* enhancer sequences with mutated Opa or Run binding sites (i.e. *sogD_ΔOpa*, *sogD_ΔOpa4* and s*ogD_ΔRun*) were chemically synthesized (GenScript) and ligated into the *eve2-promoter-MS2.yellow-attB* vector using standard cloning methods as previously described (*Koromila and Stathopoulos, 2019*; *Koromila and Stathopoulos, 2017*). Site-directed transgenesis was carried out using a *D. melanogaster* stock containing attP insertion site at position ZH-86Fb (Bloomington stock #23648). Two constructs with either five Opa sites or a single site mutated (*sogD_ΔOpa* and *sogD_ΔOpa4*, respectively) were generated to check *sog_Distal*'s expression levels upon different levels of the activator Opa at nc14A and later. Mutated site sequences and their wild-type equivalent fragments are listed below:

- *sogD_ΔRun*: tgcggtt >tAcgAtt
- *sogD_ΔOpa:* aaatt**cccacca** >aTGttcTcacca (1), gcgcc**cctttta** >gcgcATcttttta (2), cttttt**cccacgc** >cttttcTcaTTc (3), caacg**cccgcca** >caacgcATgcca (4), gaata**cccacga** >ATGtacTcacga (5)
- *sog_DistalΔOpa4*: caacg**cccgcca** >caacgcATgcca

## In situ hybridizations, immunohistochemistry, and image processing

To prepare fixed samples, standard protocols were used for 2–4 hr embryo collection, fixing, and staining (T = 23°C). Samples were collected, stained, and processed in parallel and confocal microscope images were taken with identical settings. Specifically, enzymatic in situ hybridizations were performed with antisense RNA probes labeled with digoxigenin-, biotin- or FITC-UTP to detect reporter or endogenous gene expression. *sna, hb, sog* (both full-length and intronic), and *opa* intronic riboprobes were used for multiplex fluorescent in situ hybridization (FISH).

For immunohistochemistry, anti-Opa (rabbit; this study and *Mendoza-García et al., 2017*) and anti-Zld (rabbit; this study) antibodies were used at 1:200 dilution. The anti-Opa antibody was raised in two rabbits. The immunizing antigen was a polypeptide extending from amino acid 125 to 507 of Opa. The rabbits were labeled E990 and E992. E990 antibody and respective pre-immune serum (used as control) were used previously (*Mendoza-García et al., 2017*). For production of anti-Zld antibody, an ~1 kB fragment, corresponding to aa M1240-Y1596 of the Zld peptide (junction sequences: gcgtggatccATGCAGCACCATCAG and CTCTACTGAATGAGTcgactcgagc), was amplified and cloned into the BamHi and SalI sites of pGEX-4T-1 (GE Healthcare/Millipore Sigma) and used to immunize rabbits (Pocono Farms). The nuclear staining identified by anti-Zld antibody in wt embryos (*Figure 1E*) is lost in *sh_zld* embryos, demonstrating specificity.

For the quantification of the anti-Opa antibody staining in wt, *sh_zld* and *sh_opa* embryos (*Figure 3A*), we selected the region of expected Opa expression (grey rectangular box) in ImageJ/ Fiji and used a combination of this software's available tools: 1) the Calibration bar (Analyze >Tools >Calibration bar) was used in order to establish the intensity range of each fluorescent image's selected area, and 2) Measure (Analyze >Measure) was used to make intensity measurements (mean) of each selected area. Images were taken under the same settings, 26–30 Z-sections through the nuclear layer at 0.5 µm intervals, on a Zeiss LSM 880 laser-scanning microscope using a 20x air lens for fixed embryos.

## Live imaging, data acquisition and analysis

In order to monitor expression of the various *sog_Distal* reporters described above in live embryos, virgin females containing RFP-Nucleoporin (Nup) and MCP-GFP (i.e. *yw; RFP-Nup; MCP-GFP*) were crossed with males containing the s*ogD_MS2* reporter variants (i.e. wt or *ΔOpa*). Live confocal imaging on a Zeiss LSM 880 microscope as well as imaging optimization, segmentation, and data quantification were conducted as previously described (*Koromila and Stathopoulos, 2019*).

For quantification purposes, the number of active nuclei, defined by counting dots (x-axis), was plotted against relative DV axis embryo-width (EW) position (y-axis), as analyzed from representative stills. In *Figure 1B* plots, black dotted traces overlay raw counts of MS2-MCP active nuclei-dots (bins represent minimum of four dots) detected throughout nc14 embryos containing indicated constructs after projection of scans of individual timepoints were collapsed along the anterior-posterior (AP) axis. Dots were then counted and binned across the DV axis (EW) (for details see: *Koromila and Stathopoulos, 2019*). The black line for either *sog_Distal* wild-type or mutant reporter constructs represents normalization after applying a smoothing curve. Such data were obtained and averaged for three representative videos (n = 3) of each genotype.

## Genome-wide RNA-sequencing and data analyses

Following total RNA isolation from control and *sh_opa* single embryos, RNA was quality controlled and quantified using a Bioanalyzer. Next, poly-A purified samples were converted to cDNA and high-throughput sequencing was performed to generate Illumina sequencing data by Fulgent Genetics. RNA-seq libraries were constructed using NEBNext Ultra II RNA Library Prep Kit for Illumina (NEB #E7770) following the manufacturer's instructions. Resulting DNA fragments were end-repaired, dA tailed and ligated to NEBNext hairpin adaptors (NEB #E7335). After ligation, adaptors were converted to the 'Y' shape by treating with USER enzyme and DNA fragments were size selected using Agencourt AMPure XP beads (Beckman Coulter #A63880) to generate fragment sizes between 250 and 350 bp. Adaptor-ligated DNA was PCR amplified followed by AMPure XP bead clean up. Libraries were quantified with Qubit dsDNA HS Kit (ThermoFisher Scientific #Q32854) and the size distribution was confirmed with High Sensitivity DNA Kit for Bioanalyzer (Agilent Technologies #5067–4626).

The collected raw FASTQ data files were trimmed to 40 bp paired-end reads for downstream analysis. To ensure sample identity, reads were first mapped to Gal4-VP16 sequence (Addgene #71728) associated with MTD-Gal4 (*Petrella et al., 2007*) using the BWA aligner. The read count statistics are included in *Figure 3—source data 1*. Sequencing reads were then aligned to *Drosophila* reference genome assembly (UCSC dm6) using TopHat2 (*Kim et al., 2013*) using no coverage search to speed up the process, and default settings for other parameters. Bam format of data alignment files and GTF format of the UCSC dm6 reference gene file were loaded to Cuffquant module of Cufflinks (*Trapnell et al., 2012*) to quantify gene expression. Differential expression analysis was performed using Cuffdiff module of Cufflinks with default parameters, and FPKM (Fragments Per Kilobase of transcript per Million mapped reads) values were normalized by the geometric method that Cuffdiff recommends. To identify a gene or transcript as differentially expressed, Cuffdiff tests the observed log fold change in its expression against the null hypothesis of no change (i.e., that the true log fold change = 0). Because measurement error, technical variability, and cross-replicate biological variability might result in an observed log fold change that is nonzero even if the gene/transcript is not differentially expressed, Cuffdiff also assesses the significance of each comparison. A gene is considered significantly affected if the adjusted p-value (q-value) is less than 0.05 between two groups. Consistency of differentially expressed genes across replicate samples was assessed by visualizing gene z-score values in a heatmap (generated using the R heatmap.2 function). The Z-score value is calculated as sample FPKM value minus population mean, divided by population standard deviation. In addition, a volcano plot was generated to show gene log2(fold change of expression) vs. -log2(adjusted p-value of change).

## ChIP-seq procedure

Opa-ChIP was performed as described previously (*Mendoza-García et al., 2017*) using chromatin prepared from 100 mg of pooled collections of 3 hr (2.5–3.5 hr collection) and 4 hr (3.5–4.5 hr collection) *y w[67c23]* embryos with 10 ug affinity-purified anti-Opa antibodies from two different rabbits. Control ChIP-seq libraries were generated from input chromatin as well as from a ChIP assay done with preimmune serum from one of the two rabbits. The precipitated DNA fragments were ligated with adaptors and amplified by 10 cycles of PCR using NEBNext Ultra II DNAlibrary Prep Kit for Illumina (NEB) to prepare libraries for DNA sequence determination using Illumina HiSeq2500 and single-end reads of 50 bp. The libraries were quantified by Qubit and Bio-Analyzer (Agilent Bioanalyzer 2100).

## ChIPseq data processing

The raw fastq data (50 bp single-end) for Opa ChIP-seq libraries were generated from the Illumina HiSeq2500 platform. The raw data for Zld ChIP-seq (GSM763061/GSM763062) and histone H3K4me1/H3K4me3 (GSE58935) were downloaded from the Gene Expression Omnibus (GEO) database. Trimmomatic-0.38 tool (*Bolger et al., 2014*) was used to remove Illumina adapter sequence before alignment to the *Drosophila* dm6 reference genome assembly with the Bowtie2 alignment program (*Langmead and Salzberg, 2012*). Alignment BAM files were subject to further sorting and duplicate removal using the Samtools package (*Li et al., 2009*). Reads mapped to chr2L, chr2R, chr3L, chr3R, chr4, chrX were kept and biological replicate BAM files were merged for downstream analysis. ChIP-seq signal trace files were generated using the bamCoverage function of deepTools (*Ramírez et al., 2014*), with RPKM normalization and 10 bp for the genomic bin size.

Both IP and input data were used for ChIP-seq peak calling. For calling of transcription factor binding sites, a workflow using bdgcmp and bdgpeakcall modules of the MACS2 peak caller (*Zhang et al., 2008*) was utilized. Peak calling was performed using merged replicate ChIP data (to improve the sensitivity of the peak calling by increasing the depth of read coverage) against input data (a proxy for genomic background). As noted, visual inspection of signal traces of both preimmune negative control data and genomic input data showed a clean background, thus mapped reads were merged to serve as background for ChIP-seq peak calling. Genomic regions with q-values of less than $10^{-5}$ were defined as ChIP-seq peak regions. To understand overlapping of Opa and Zld binding sites across the genome, Opa and Zld peak regions were combined and overlapping peaks were merged. Combined regions that overlapped both Opa and Zld peaks were defined as Opa-Zld overlap regions; regions overlapping with either Opa or Zld peaks were defined as Opa-only and Zld-only regions respectively. Further de novo motif analysis was performed on different ChIP-Seq regions using the HOMER program (*Heinz et al., 2010*) with default parameters and with options -size 200 and -mask. The most enriched de novo motifs identified from Opa ChIP-seq peaks and from Zld ChIP-seq peaks were queried against the Opa-Zld overlap, Opa-only and Zld-only regions for comparison and for generating aggregation plots. Average ATAC-seq signals around different ChIP-seq regions were also calculated using the annotatePeaks.pl module of HOMER, with the -size 4000 -hist 10 options used for aggregation plots. Also different ChIP-seq regions were annotated and linked to the nearest gene transcription start sites. Functional gene annotation was performed using DAVID v6.7 (https://david.ncifcrf.gov/home.jspcitation). In addition, computeMatrix and plotHeatmap modules of deepTools were used to calculate and plot normalized histone mark and ATAC-seq signal intensities around different ChIP-seq regions. DNA sequence logos were plotted using the seqLogo R package. Region overlap analysis was performed using an online tool (http://bioinformatics.psb.ugent.be/webtools/Venn/). Unless noted otherwise, R was used to generate plots. For this and all subsequent data presented using heatmaps, the first sample in the heatmap was used for sorting the genomic regions based on descending order of mean signal value per region; all other comparison samples were plotted using the same order determined by the first sample.

## Single-embryo ATAC-seq procedure

Embryos were collected on agar plates from females of the following genotypes: wild-type/control (i.e *y w* females crossed to *sh_opa* males), mutant (i.e. *opa1* and *MTD-Gal4*, *sh_opa* or *sh_zld*), or ectopically-expressing *opa* (i.e. *MTD-Gal4*, *UAS-opa*). Individual embryos were selected from plates, and nuclear morphology was observed live under a compound microscope at 20x magnification. Temperature for sample collection was maintained at 26°C within an incubator to minimize variation in staging. Under these conditions, cell cycling timing was indistinguishable between genotypes. The staging of the samples started at 3 min intervals from the onset of anaphase of the previous cell cycle. Each embryo was hand-selected and hand-dechorionated for the analysis. Prepared libraries were subject to either paired-end [wt (at nc14B and nc14D), *UAS-opa* (at nc14B), *opa1* (at nc14D) and *sh_opa* (at nc14D); average of three single embryo replicates] or single-end sequencing (wt and *sh_zld*; average of one nc14B and one nc14D samples per timepoint as only these data passed quality control after sequencing) of 50 bp reads, using an Illumina HiSeq2500 platform. Fragmentation and amplification of single-embryo ATAC-seq libraries were performed essentially as described previously (*Blythe and Wieschaus, 2016b*; *Buenrostro et al., 2015*). Single embryos embryos were

collected at nc14+20 min for nc14B and nc14+45 min for nc14D (T = 26°C). Developmental progression of individual embryos was monitored under a microscope, and embryos harvested at the indicated times ± 2 min (T = 23°C).

## ATAC-seq processing, mapping and peak calling

ATAC-seq reads were trimmed and filtered using Trimmomatic (version 0.33) (*Bolger et al., 2014*) and cutadapt (version 1.15) (*Martin, 2011*). The first 30 bp from each read were mapped using Bowtie2 (version 2.1.0, parameters: `-end-to-end -very-sensitive -no-mixed -no-discordant -q -phred33 -I 10 -X 700`).

HOMER (version 4.7, parameters: -localSize 50000 -minDist 50 -size 150 -fragLength 0) (*Heinz et al., 2010*) was used to call ATAC peaks. The peaks that overlap ENCODE 'blacklist regions' (*Amemiya et al., 2019*) were removed.

For the individual loci ATAC-seq data that are depicted in *Figure 5C,D,G* and *Figure 5—figure supplement 1B, C, E and F*, mapped reads were normalized similarly to a published method for better visualization (*Blythe and Wieschaus, 2016b*). First, to define the background, 150 bp peaks were called from the original data using HOMER (-localSize 50000 -minDist 50 -size 150 -fragLength 0) to capture most of the non-background regions. These 150 bp peaks were extended from the center to form 20000 bp 'signal zones'. Outside these signal zones are 'background zones'. Next, to sample the background noise, 100000 150 bp random regions were generated. Those 150 bp random regions that completely fell into the 'background zones' were regarded as 'background regions'. The mean and standard deviation for the background noise were calculated from positive RPM scores of each nucleotide in these regions (ypbkg) based on log-normal distribution. Finally, RPM scores for the whole genome were centered and scaled based on the mean and standard deviation calculated, using one as pseudocount:

$$log_2(y_{norm} + pseudocount) = \frac{log_2 y - \overline{log_2 y_{pbkg}}}{std(log_2 y_{pbkg})}$$

## Integrative analysis of multi-omics data

ChIP-seq peak-associated genes and RNA-seq differentially expressed genes were subjected to overlapping count calculation, and the results were presented in a bar plot. To understand changes of chromatin accessibility surrounding transcription factor binding sites, ATAC-seq signals (average from three single embryo biological replicates; except for wt and *sh_zld* singled-end ATAC-seq data, as described above) within 1 kb genomic bins surrounding different categories of ChIP-seq regions were calculated, and presented in a box plot. For comparison, ATAC-seq signals surrounding ATAC-seq peak regions were also calculated and presented in a box plot (*Figure 5—figure supplement 1D*).

## ATAC-seq differential peaks

We grouped mutant samples and control samples into two separate groups and merged all the aligned reads separately. Peaks were called for the two merged samples using the method described above. We were particularly interested in the peaks that were less accessible in *sh_opa* embryos (n = 3). Therefore, the peaks called from the merged control sample (n = 3), were converted into broad peaks by extending 200 bp upstream and downstream and merging overlapped ones. These broad peaks were used as candidate input and differential peaks called from these processed datasets using the getDifferentialPeaks function (parameters: -size 200 F 2) from HOMER (*Heinz et al., 2010*).

## Nucleosome signature analysis

From the broad peaks called from merged *UAS-opa* and control ATAC-seq samples at nc14B using the method described above, we called nucleosome locations using NucleoATAC based on fragment size and using default parameters (*Figure 5E,F* and *Figure 5—figure supplement 1G, H*; *Schep et al., 2015*). The peaks that had at least one nucleosome called by NucleoATAC were selected for downstream analyses. Genome motif scanning (fimo pipeline) using an Opa binding site consensus (JASPAR MA0456.1) revealed 25921 matches across the genome. These matches were further divided into 4481 'bound' matched positions that overlap with Opa (3 hr) ChIP-seq peaks

and 21440 'unbound' ones that do not. Similarly, 3276 'bound' and 22645 'unbound' motif positions were also derived from those 25921 matches for Opa (4 hr) ChIP-seq peaks. For each of these four categories, matched motif positions that overlapped with the broad ATAC-seq called peaks (either *UAS-opa* or control samples) that also had at least one nucleosome called were identified. The distances between each motif location and its nearest nucleosome were recorded and plotted.

## Acknowledgements

We thank Chris Rushlow and Deborah Hursh for sharing fly stocks, Igor Antoshechkin and Henry Amrhein at the Millard and Muriel Jacobs Genetics and Genomics Laboratory at the California Institute of Technology for sequencing support, the lab of Josh Dubnau for assistance with Bioanalyzer samples, David Carlson and the Institute for Advanced Computational Science at the Stony Brook University, and Susie Newcomb, Leslie Dunipace and Frank Macabenta for assistance with experiments and comments on the manuscript. This study was supported by funding from NIH R35GM118146 and R03HD097535 to AS, the Bioinformatics Resource Center at the Beckman Institute of Caltech to FG and LP, and the Stony Brook University College of Arts and Sciences to JPG.

## Additional information

### Funding

| Funder | Grant reference number | Author |
| --- | --- | --- |
| National Institute of General Medical Sciences | R35GM118146 | Angelike Stathopoulos |
| Eunice Kennedy Shriver National Institute of Child Health and Human Development | R03HD097535 | Angelike Stathopoulos |
| Bioinformatics Resource Center at the Beckman Institute of Caltech | | Fan Gao Lior Pachter |
| Stony Brook University College of Arts and Sciences | | J Peter Gergen |

The funders had no role in study design, data collection and interpretation, or the decision to submit the work for publication.

### Author contributions

Theodora Koromila, Conceived the project and planned the experimental approach, performed wet experiments except ChIP-seq, oversaw computational approach, carried out quantitative analysis of imaging data, analyzed data, wrote manuscript with input and editing help from FG, YI, PH, LP and JPG.; Fan Gao, Oversaw computational approach, performed all computational analysis except normalization of ATAC-seq data for visualization of individual loci, ATAC-seq peak calling, and nucleosome signature, analyzed data, gave input and editing help for writing the manuscript.; Yasuno Iwasaki, Performed ChIP-seq experiments with support of the Caltech genomics core, conducted an initial, independent analysis of the Opa-ChIP-seq data that first identified the new 7 bp consensus binding motif for Opa, gave input and editing help for writing the manuscript.; Peng He, Oversaw computational approach, conducted normalization of ATAC-seq data for visualization of individual loci, ATAC-seq peak calling, and nucleosome signature, analyzed data, gave input and editing help for writing the manuscript.; Lior Pachter, J Peter Gergen, Gave input and editing help for writing the manuscript.; Angelike Stathopoulos, Conceived the project and planned the experimental approach, directed the project, analyzed data, wrote manuscript with input and editing help from FG, YI, PH, LP and JPG.

## Author ORCIDs
Theodora Koromila (iD) https://orcid.org/0000-0001-5504-1369
Peng He (iD) https://orcid.org/0000-0002-2457-3554
Angelike Stathopoulos (iD) https://orcid.org/0000-0001-6597-2036

## Decision letter and Author response
Decision letter https://doi.org/10.7554/eLife.59610.sa1
Author response https://doi.org/10.7554/eLife.59610.sa2

## Additional files

### Supplementary files
• Transparent reporting form

### Data availability

GEO accession number SuperSeries GSE153329. SubSeries: ChIP-seq and singled-end ATAC-seq (GSE140722), and RNA-seq and paired-end ATAC-seq data access (GSE153328). The codes for RNA-seq, Opa ChIP-seq and ATAC-seq processing (alignment and peak calling) were uploaded to github: https://github.com/caltech-bioinformatics-resource-center/Stathopoulos_Lab (copy archived at https://github.com/elifesciences-publications/Stathopoulos_Lab).

The following datasets were generated:

| Author(s) | Year | Dataset title | Dataset URL | Database and Identifier |
|---|---|---|---|---|
| Koromila T, Gao F, Iwasaki Y, He P, Pachter L, Gergen P, Stathopoulos A | 2019 | ChIP-seq and singled-end ATAC-seq | http://www.ncbi.nlm.nih.gov/geo/query/acc.cgi?acc=GSE140722 | NCBI Gene Expression Omnibus, GSE140722 |
| Koromila T, Gao F, Iwasaki Y, He P, Pachter L, Gergen P, Stathopoulos A | 2020 | RNA-seq and paired-end ATAC-seq data | http://www.ncbi.nlm.nih.gov/geo/query/acc.cgi?acc=GSE153328 | NCBI Gene Expression Omnibus, GSE153328 |

The following previously published datasets were used:

| Author(s) | Year | Dataset title | Dataset URL | Database and Identifier |
|---|---|---|---|---|
| Harrison MM, Li X, Kaplan T, Botchan MR, Eisen MB | 2011 | Zelda binding in the early Drosophila melanogaster embryo marks regions subsequently activated at the maternal-to-zygotic transition | https://www.ncbi.nlm.nih.gov/geo/query/acc.cgi?acc=GSM763061 | NCBI Gene Expression Omnibus, GSM763061 |
| Harrison MM, Li X, Kaplan T, Botchan MR, Eisen MB | 2011 | Zelda binding in the early Drosophila melanogaster embryo marks regions subsequently activated at the maternal-to-zygotic transition | https://www.ncbi.nlm.nih.gov/geo/query/acc.cgi?acc=GSM763062 | NCBI Gene Expression Omnibus, GSM763062 |

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
