## [Decision Letter]

[Editors’ note: the authors submitted for reconsideration following the decision after peer review. What follows is the decision letter after the first round of review.]

Thank you for submitting your work entitled "Odd-paired is a late-acting pioneer factor coordinating with Zelda to broadly regulate gene expression in early embryos" for consideration by *eLife*. Your article has been reviewed by three peer reviewers, and the evaluation has been overseen by a Reviewing Editor and a Senior Editor. The following individuals involved in review of your submission have agreed to reveal their identity: Erik Clark (Reviewer #3).

Our decision has been reached after consultation between the reviewers. Based on these discussions and the individual reviews below, we regret to inform you that your work can not be considered for publication in *eLife* in its present form. While all reviewers agreed that this work was of general interest, they all raised a number of concerns that require substantial additional experimentation that we think is outside the presently required 2 month time window. Given the general interest of the study, we do encourage you to consider these requested experiments. Should you be able to address these concerns we would be interested in seeing a substantially revised version of the paper submitted as a new submission.

Reviewer #1:

Kormila et al. build on prior publications demonstrating that Runt activity at the sog distal enhancer transitions from a repressor to an activator by identifying Odd-paired (Opa)-binding to this regulatory element. They use live-embryo imaging to show that Opa-binding sites are required for late activation from this regulatory element and, using ChIP-seq and ATAC-seq analysis, they argue that Opa is a pioneer factor essential for a late wave of zygotic genome activation. While interesting, the genomic data as presented are somewhat superficially analyzed and do not support a clear role for Opa in driving chromatin accessibility, a defining feature of pioneer transcription factors.

1) The ChIP data would benefit from additional analysis. To determine biologically meaningful differences between the various ChIP peaks, the authors should analyze the relative peak heights for the 16,085 peaks. For example, are there any differences in relative peak heights for the peaks bound by Opa alone, Opa and Zld, or Zld alone? In Figure 3—figure supplement 1, it appears that the peaks uniquely bound by Opa at a single developmental time point (3h vs. 4h) are lower than the peaks shared between time points (assuming color indicates relative peak heights). If this is the case, these differences in binding may actually reflect variability in ChIP efficiencies for these lower peaks rather than meaningful biological changes. Relative peak heights can then be used in the analysis of the ATAC-seq to determine if those regions with the highest ChIP signal are correlated with accessibility.

2) For the ATAC-seq data statistical methods (i.e. DESeq2 or edgeR) should be used to identify significant changes in accessibility between wild-type and shRNA (*opa* or *zld*) embryos. How many regions lose accessibility overall? Once significant changes are identified, the overlap between these regions and binding of Zld and Opa can be directly tested. Are the regions that change in accessibility upon knockdown those with the highest ChIP-seq signal for the identified factor?

3) Better confirmation needs to be demonstrated for the shRNA knockdown. Given that the authors use anti-Opa and anti-Zld antibodies in Figure 1C to demonstrate protein expression this should be used on shRNA embryos to demonstrate protein knockdown and not just a decrease in the amount of RNA. Alternatively, western blots on bulk embryos should be used to demonstrate a decrease in levels of the protein products of the genes being targeted. In addition, because the Zld antibodies used in this manuscript have not been previously published information regarding antibody production and a demonstration of specificity needs to be included in the Materials and methods. Also, the staging of the single embryos used for the ATAC-seq should be noted.

4) The data presented alone do not allow the conclusion that Opa is a pioneer factor and thus the title for Figure 5 and some of the conclusions must be softened or additional data provided. There is no data presented that demonstrates that Opa establishes accessibility by accessing previously nucleosome-bound regions. In addition, there is a limited demonstration that the accessibility that is lost in an *opa* mutant has direct effects on either the ability of additional transcription factors to bind or gene expression. The prior analysis of Runt binding to the sog distal enhancer provides a mechanism for them to directly test the requirement for Opa-mediated accessibility in facilitating Runt binding. ChIP-qPCR for Runt on the sog distal enhancer reporter with the mutated Opa binding sites (Figure 1) and/or in the shRNAi knockdown would begin to address whether Opa has these additional defining features of a pioneer factor. In addition, the global effect of Opa-mediated accessibility on gene expression could be analyzed by RNA-sequencing.

5) There are a number of citations that should be added to the manuscript.

Kwasnieski et al., 2019 should be included with Ali-Murthy and Lott.

Xu et al., 2014 and Yanez-Cuna et al., 2014 should be included with Harrison, Liang and Nien.

Sun et al. Genome Research 2015 should be included with Schulz.

Some mention of Li et al., 2008 and the included demonstration that A/P factors bind D/V enhancers should be included.

Papers from which ChIP-seq data sets were analyzed should be cited.

Harrison et al., 2010 should be substituted for Schulz et al., 2015.

Reviewer #2:

Based on the finding that the transcription factor *opa*, primarily known for its function during A/P segmentation, regulates the expression of a D/V enhancer (sog) at late stages, the authors question the role of *opa* at a whole genome scale. By performing *opa* ChIP-seq experiments as well as ATAC-seq in wt and embryos depleted for *opa*, the authors identify a new set of cis-regulatory regions bound by *opa* and whose accessibility is *opa*-dependent. By comparing these accessibility profiles to Zelda-dependent ones, the authors propose that *opa* acts a pioneer factor to control the timing of gene expression.

While the finding that *opa* controls accessibility and could act as a general timing factor during MBT that acts subsequently to Zld-mediated activation is novel and exciting, I have some reservations concerning the evidence supporting this finding. While the overall manuscript is promising, there are some issues with precision and rigor the authors should address. If these revisions are made, I would recommend this work for publication in *eLife*.

General comments:

The title is “Odd-paired is a late acting pioneer factor…”. The defining properties of a pioneer factor are: a) protein binding to nucleosomal DNA; b) retention of the protein during mitosis; c) a general requirement for establishing/maintaining accessibility of the genome; d) cis-regulatory element binding prior to target gene activation; and e) a general function in reprogramming. While not all of these properties strictly need to be met to declare a TF a pioneer factor, the current manuscript only demonstrates that *opa* is necessary for accessibility. The title requires nuance, and the authors should discuss similarities and differences between *opa* and other pioneer factors within the body of the manuscript.

The claim that *opa* acts as a timing factor is not fully supported by the data. Zelda-mediated activation timing has been demonstrated by accelerating activation with extra Zld sites or delaying it via deletion of Zld sites (Foo et al., Crocker et al., Dufourt et al., Yamada et al., etc.). To support *opa* regulation of timing, similar experiments would strengthen this claim immensely. Alternatively, the authors could drive maternal expression of *opa* and examine the change in temporal behavior on their existing Sog-MS2 transgene.

Specific comments:

1) Analysis of *opa* regulation of SogD-MS2 transgene

– In Figure 1A, a hole could be indicative of an unhealthy embryo. Moreover, Video 1 presented in the supplementary data does not correspond to these still images. The authors should provide images from a healthy embryo and show the corresponding video.

– In Figure 1, the number of videos should be indicated. Their quantification in terms of % of activation should be moved from the supplementary data to the main figure.

-In Figure 1—figure supplement 1, the authors should add the corresponding sog_unmutated control panels and the quantification of each genotype in terms of % activation.

– The images in Figure 1B suggest that reporter expression abnormally persists in the mesoderm of *sogD_ΔOpa* at nc14a. If true, the authors should comment on this result. Did the authors check that the *opa* mutations do not affect twi or sna binding sites?

2) ChIP-seq comparison of Zelda vs. *opa*

– The reference for the Zelda ChIP-seq data should be indicated (subsection “Assay of overrepresented sites associated with Opa ChIP-seq peaks”). In particular, are *opa* ChIP 3h samples compared to an equivalent Zelda 3h dataset (and *opa* 4h to Zelda 4h)?

3) Investigating the role of *opa* vs. Zelda for chromatin accessibility

– The authors did not justify the need for single embryo ATAC-seq. Single embryo studies are most useful if the exact developmental timing is known. If the authors did precisely time their experiment, this value should be reported with the data.

– The authors should explicitly state that they used the same Gal4 driver for RNAi Zelda and RNAi *opa*. (subsection “Global changes in chromatin accessibility result upon knock-down of Opa”)

– To underline that *opa*-bound enhancers are active later than Zelda-only enhancers, the authors should show RNA-seq tracks for the genes exemplified in Figure 3 A-H

– The authors mention that the *opa1* mutant phenotype is comparable to that of *opa* RNAi embryos, but this is not shown or cited from another publication. Figure Sup4 should be complemented by similar FISH/immunolabeling in *opa* RNAi embryos.

– The RNAi stock used to deplete *opa* is not homozygous viable (given the stock described in the Materials and methods), possibly suggesting off-target effects of the RNAi. To circumvent this possibility, the authors should perform single embryo ATAC-seq on *opa1* mutants. If the accessibility results are similarly affected to those in *opa* RNAi then it would strengthen their conclusions.

– The author seems to have performed new ATAC-seq experiments on RNAi Zld, but the driver employed needs to be specified. Could the authors compare their results with published single embryo ATAC-seq in Zelda mutants (Hannon, 2017)?

– To be rigorous, the authors should have used a RNAi-white crossed with the same MTD-gal4 line as a control and not WT embryos. However, I think this experiment is less important than performing ATAC-seq in *opa1* mutants.

4) *opa* and chromatin accessibility

– Figure 4D suggests that accessibility seems much higher for Zelda/Opa common targets than for each TF target independently. Accessibility seems also higher for *opa* peaks compared to Zelda peaks. Is this data quantitative or semi-quantitative at all, and is there a way to explain that within the text? Can these curves be statistically compared? Can the authors produce similar graph at earlier and later stages?

– The accessibility results of Figure 4 should be complemented with Zelda and *opa* ChIP-seq tracks and organized to better emphasize the 3 groups of accessibility identified by the authors.

– It would be interesting to compare the *opa* peaks that are accessible independently of Zld to the genes that remain accessible in Zld mutants (Harrison, 2010, Schultz, 2015, Hannon, 2017).

I found figures 4 and 5 to be confusing. If the central idea of Figure 4 is to show that *opa* is responsible for chromatin accessibility, the authors should only present WT vs. *opa* RNAi. Then in Figure 5, they could add the Zelda RNAi comparison.

– Subsection “Opa-only occupied peaks require Opa to support their accessibility at mid-nc14”/Figure 4G: the tracks give no information on whether Zelda binds the eve LE enhancer. The authors could add the ChIP-seq tracks as performed in Figure 4—figure supplement 2A-B to clarify this. Additionally, the loss of accessibility in Zelda RNAi at this enhancer is shown but not commented on in the text.

– Also the panels of figures mentioned in the text are confusing and need revising. For example: Figure 3 is mentioned even though it does not show any accessibility data. Figure 4—figure supplement 1F does not exist.

– Subsection “Opa-only occupied peaks require Opa to support their accessibility at mid-nc14” paragraph three: The authors use the term *opa1* mutant, when in fact they are looking at *opa* RNAi-mediated transcript depletion, which is significantly different.

Reviewer #3:

Opa is a zinc finger TF that is expressed broadly in *Drosophila* embryos during the latter part of cellularisation, gastrulation, and GBE. Koromila and colleagues use Opa ChIP-seq along with ATAC-seq from wt and *opa* RNAi embryos to show that Opa binds to thousands of regions across the *Drosophila* genome and is required for chromatin accessibility at many of them. The case study of the sog enhancer *sog_Distal* links these phenomena with effects on gene expression: mutating Opa binding sites present within this enhancer reduces the expression of a reporter gene at timepoints when Opa is expressed. The paper argues that Opa is an important pioneer factor that ushers in a second major wave of zygotic gene expression, separate and later than the one brought about by a different (and more extensively studied) zinc finger TF, Zelda. This is an important discovery, and along with a recent study from the Blythe lab that reaches similar conclusions, this paper will surely cause many researchers to investigate whether and how Opa is regulating their gene/developmental process of interest, in *Drosophila* and beyond.

I do have some concerns about the staging of the embryos. In particular, the central claim of the paper is that Opa coordinates with Zelda in regulating gene expression, because it binds to many of the same genomic regions, and for some of these regions, *opa* knockdown and zld knockdown both affect accessibility. Simultaneous Opa+Zld binding is inferred by comparing Opa ChIP-seq peaks to a published Zld ChIP-seq dataset. However, the Zld dataset uses embryonic stages (nc13 and early nc14) from before Opa is expressed, meaning that the possibilities of simultaneous binding vs. sequential binding cannot be distinguished. If possible, I would have the authors re-run their analysis using the "late nc14" dataset from the same paper instead, which seems a more appropriate comparison for their purposes. As an optional extension, explicit comparison between the early and late Zld datasets could also give interesting hints as to the nature of any Opa/Zld interaction – for example when Opa starts binding to the genome in late nc14, does this cause Zld binding at these loci to increase or reduce, relative to other Zld-bound loci where Opa is absent?

[Editors’ note: further revisions were suggested prior to acceptance, as described below.]

Thank you for submitting your article "Odd-paired is a pioneer-like factor that coordinates with Zelda to control gene expression in embryos" for consideration by *eLife*. Your article has been reviewed by three peer reviewers, and the evaluation has been overseen by a Reviewing Editor (Oliver Hobert) and Kevin Struhl as the Senior Editor. The following individual involved in review of your submission has agreed to reveal their identity: Erik Clark (Reviewer #1).

The reviewers greatly appreciate the revisions, compared to an earlier version of the manuscript that you had submitted and all agree that the study is now almost ready for acceptance. A few minor editorial issues remain and are listed below in the reviewers comments. Once those are fixed we expect the manuscript to be acceptable for publication.

Reviewer #1:

The manuscript has been significantly improved by its revisions. My key concerns have both been addressed by the authors – by 1) clarifying that the onset of Opa expression happens at nc14b and by 2) additionally comparing their Opa ChIP-seq dataset to a later Zld dataset, as requested. The authors have also carried out a considerable amount of new experiments and analysis (despite their current lab shutdown), and these new data strengthen their proposal that Opa is a key timing factor in the early *Drosophila* embryo. The authors' responses to my original comments are reasonable. I don't think any further experiments or analyses are necessary but the text could be further edited for clarity. I felt that the first half of the paper read well, but the second half of the paper lacked the same degree of polish and was sometimes hard to follow.

Reviewer #2:

This is a much improved manuscript that has worked to address many of the significant concerns from the prior submission. The additional data strengthen the conclusions of the manuscript. Furthermore, the focus on the Opa (3h) data successfully streamline the manuscript such that the focus remains on the conclusions most robustly supported by the data. We have only minor issues that should be addressed prior to publication.

1) The additional data with the ATAC-seq upon precocious expression of Opa and the analysis on nucleosome distance reported in Figure 5 are nearly identical to experiments published by Soluri et al., 2020. As such, this publication must be briefly discussed and cited. It is gratifying to show that these results are robust across laboratories and acknowledgement of this fact does not decrease the impact of this publication. We feel that this addition to the manuscript is necessary for acceptance.

2) Figure 3D is confusing as it appears that only a very small handful of genes are bound by Opa. This is obviously not the case as shown in Figure 3—figure supplement 1F, but the authors should consider a different way of highlighting specific genes. As it is, the black dots are labelled "Opa only" but these are clearly only a small fraction of the Opa only bound genes in this volcano plot. Similarly for the yellow Opa/Zld genes. It could be useful to report on the plot the % of down- and up-regulated genes that are proximal to an Opa binding site.

3) For the various heat maps, the order in which peaks are ranked should be clearly indicated. For example, in Figure 4F it is unclear whether each heat map is ranked separately or whether one can compare across heat maps. Similarly, the method for ranking peaks should be provided for Figure 5A and B.

4) There are a few typos/ formatting errors.

- Results paragraph three, the authors state that Opa expression is reduced at nc14c for a mutant reporter. It is not clear whether the authors mean Opa expression in this case.

- In many of the figures, the symbol for α in antibody staining is an "a".

- The authors state that there is a "significant increase in chromatin accessibility across Ope-bound regions (Figure 5A)." While these data are compelling, the word significant implies some sort of statistical analysis. If such an analysis was performed this should be reported. Otherwise, a change in word choice should suffice.

- The citation (Blythe, 2016) in paragraph three of the Discussion should presumably be Blythe and Wieschaus, 2016.

- In the legend for Figure 1I “*Mus musculus*" should have the genus name capitalized and should be italicized.

- The inclusion of Su(H) in Figure 6E is confusing and should be removed.

Reviewer #3:

The manuscript has been significantly improved.

The authors have answered the vast majority of my concerns and requests.

They have conducted extra experiments (such as paired-end ATAC-seq and single embryo RNA-seq in control and Opa RNAi backgrounds).

The notion of a “pioneer factor” was more thoroughly discussed. The new data/analysis concerning Opa-driven nucleosome signatures supports the notion that Opa exhibits the properties of a pioneer factor.

The text has been extensively revised, as well as the organization of the figures.

Given this pandemic period, revisions must not have been simple to perform, and I therefore highly congratulate the authors for their work. The revised manuscript is now suitable for publication.

---

## [Author Response]

[Editors’ note: the authors resubmitted a revised version of the paper for consideration. What follows is the authors’ response to the first round of review.]

Reviewer #1:1) The ChIP data would benefit from additional analysis. To determine biologically meaningful differences between the various ChIP peaks, the authors should analyze the relative peak heights for the 16,085 peaks. For example, are there any differences in relative peak heights for the peaks bound by Opa alone, Opa and Zld, or Zld alone? In Figure 3—figure supplement 1, it appears that the peaks uniquely bound by Opa at a single developmental time point (3h vs. 4h) are lower than the peaks shared between time points (assuming color indicates relative peak heights). If this is the case, these differences in binding may actually reflect variability in ChIP efficiencies for these lower peaks rather than meaningful biological changes.

We use heat maps to analyze relative peaks heights for the classes indicated (i.e. Opa alone, Opa and Zld, or Zld alone); yes, color indicates relative peak heights. Regarding the data in question (currently shown in Figure 4—figure supplement 1), it is theoretically possible that the difference between Opa_early only and Opa-late only peaks relates to ChIP efficiencies. However, both experiments were conducted in duplicate and none of the Opa_early only or Opa-late only peaks were associated with negative control (input/preimmune ChIP).

In addition, when we examined chromatin accessibility at Opa ChIP-seq peaks present only at 3h, 4h, or continuously present in both samples, we found that the “late” occupied peaks (4h) were not forced open by increased Opa levels or closed upon loss (see Figure 6—figure supplement 1). In contrast, accessibility at Opa-bound regions identified by ChIP-seq only at the 3h timepoint (only early) or both 3h and 4h samples (i.e. present continuously, early and late) were influenced by Opa levels. This difference in behavior of peaks bound by Opa at these developmental timepoints (i.e. 3h only regions can open in response to Opa; but 4h only regions do not), in our opinion, provides support that these ChIP-seq defined binding events are meaningful and reinforces the view that Opa is influential at nc14.

For all these reasons, we believe the differences we observed in our ChIP-seq samples are biologically meaningful. Nevertheless, for most of our analyses in this study, we used the Opa (3h) dataset because we wanted to focus on nc14 and decrease the complexity/length.

Relative peak heights can then be used in the analysis of the ATAC-seq to determine if those regions with the highest ChIP signal are correlated with accessibility.

Our previous experience with ChIP-seq data coupled with careful analysis of explanatory binding sites through mutagenesis/mutant analysis, leads us to believe that peak height is not the best predictor of the “importance” of a factor for supporting gene expression through the bound enhancer element; more often than not, mutation of binding sites within enhancers associated with smaller ChIP-seq peaks have bigger effects on gene expression outputs. At least this is what we learned from such an analysis of Twist transcription factor ChIP-seq binding versus function (Ozdemir et al., 2011).

Therefore, we instead chose to use RNA-seq data (new data/Figure 3) to filter the ATAC-seq datasets. For ATAC-seq, we assayed individual embryos (i.e. precisely timed samples) at both early (nc14B) and late (nc14D) timepoints, and added ectopic expression of Opa (UAS-*opa*) experiments. These new data and analyses, we believe, has strengthened our case for Opa being a generally-acting pioneer-like factor that regulates zygotic gene expression broadly in embryos.

In particular, we added signal aggregation plots in addition to the heatmaps from pair-end nc14D *sh_opa* and nc14B *UAS_opa* and used relative peak height to claim the ATAC-seq identified changes in accessibility. Please see new Figure 5A,B and Figure 6—figure supplement 1, as well as Figure 6A,B.

2) For the ATAC-seq data statistical methods (i.e. DESeq2 or edgeR) should be used to identify significant changes in accessibility between wild-type and shRNA (opa or zld) embryos. How many regions lose accessibility overall? Once significant changes are identified, the overlap between these regions and binding of Zld and Opa can be directly tested. Are the regions that change in accessibility upon knockdown those with the highest ChIP-seq signal for the identified factor?

We tested DESeq2 and edgeR for differential peak calling and could not obtain meaningful results, suggesting intrinsic data noise was present that was not amenable to a negative binomial statistical model. We acknowledge that our initial design of sample size (3 replicates per group) for ATACseq might be insufficient to get enough statistical power to support this particular analysis. We also tried a different statistical method, the getDifferentialPeaks function (parameters: -size 200 -F 2) from HOMER (Heinz et al., 2010), which returned 600 differential peaks among the 12023 in the control sample that became closed and was also parameter-sensitive. So we decided to use the default FDR for HOMER peak caller and looked for condition-specific peaks.

ATAC-seq data from control nc14D (accessible regions=12023 HOMER peaks, 3784 not overlapped with *sh_opa* peaks) and *sh_opa* nc14D (accessible regions=12739 HOMER peaks) samples suggest that 31.5% of the regions lose chromatin accessibility in the absence of Opa at nc14D. On the other hand, there is 88.5% overlap (3301 peaks) between the control-vs-*opa1 mutant* and control-vs-*sh_opa* chromatin peaks and 75% overlap (3187 peaks) between the *sh_opa*-vs-control and *UAS_opa-vs-control* peaks (Figure 5—figure supplement 1A).

We also noticed that ATACseq signals surrounding Opa ChIP-seq peak regions are broadly altered in both *sh_opa* and *UAS_opa* (heatmaps, Figure 5A,B): decreased accessibility in *opa* RNAi (*sh_opa*) and increased accessibility upon ectopic expression of Opa (*UAS_opa*).

3) Better confirmation needs to be demonstrated for the shRNA knockdown. Given that the authors use anti-Opa and anti-Zld antibodies in Figure 1C to demonstrate protein expression this should be used on shRNA embryos to demonstrate protein knockdown and not just a decrease in the amount of RNA. Alternatively, western blots on bulk embryos should be used to demonstrate a decrease in levels of the protein products of the genes being targeted. In addition, because the Zld antibodies used in this manuscript have not been previously published information regarding antibody production and a demonstration of specificity needs to be included in the Materials and methods. Also, the staging of the single embryos used for the ATAC-seq should be noted.

We show anti-Opa stainings in *sh_opa* in Figure 3A that demonstrate decreased Opa protein levels upon RNAi knockdown.

Additional information regarding how the Zld antibody was generated and tested has been included in the Materials and methods, as the reviewer suggested.

Details regarding the staging were added in the main text as well as in the figures: all ATAC-seq data is from single embryos from two stages: either nc14B (nc14+20min) or nc14D (nc14+45min), as specified.

4) The data presented alone do not allow the conclusion that Opa is a pioneer factor and thus the title for Figure 5 and some of the conclusions must be softened or additional data provided. There is no data presented that demonstrates that Opa establishes accessibility by accessing previously nucleosome-bound regions. In addition, there is a limited demonstration that the accessibility that is lost in an opa mutant has direct effects on either the ability of additional transcription factors to bind or gene expression.

We understand the reviewer’s concern and have chosen to refer to Opa as a “pioneer-like” factor in the title, but also did add additional data to support the view that it indeed functions as a pioneer factor. New paired-end ATAC-seq data were added, which besides providing better information regarding degree of chromatin accessibility also allowed us to identify nucleosome signatures. Thes new nucleosome signature data were added to Figure 5E,F and support the view that Opa establishes accessibility by accessing previously nucleosome-bound regions.

Regarding whether loss of *opa* has direct effects on gene expression, in new Figure 3 we have added singleembryo RNA-seq data for control versus *sh_opa* mutant embryos. Furthermore, regarding whether loss of accessibility has direct effect on TF binding/gene expression, we have selected representative loci for analysis in Figures 4 and Figure 5—figure supplement 1. For example, in the case of *hb,* Opa ChIP-seq (3h) data shows binding at both the *hb_stipe* and VT38550/*hb_HG4-7* enhancers that act at nc14 as shown in Figure 4 panel A’. New ATACseq data from nc14B *opa* ectopic expression (UAS-*opa*) and nc14D *sh_opa* samples suggest Opa affects accessibility at these sequences (Figure 5—figure supplement 1E). Furthermore, expression of the *hb_stripe* (and also likely VT38550/*hb_HG4-7*) is delayed in *opa* mutants (see nc14C; Figure 6D). We also note that our study is the first, to our knowledge, to demonstrate that posterior *hb* expression is supported by VT38550/*hb_HG4-7* enhancer at nc14; there is clear Opa binding, but little Zld binding, associated with this region (Figure 4A). Collectively, these data for *hb* support the view that Opa acts at nc14, and contributes to gap gene (i.e. *hb*) expression.

Showing that the binding of another transcription factor, Zld, Bcd, Dl, and/or Twi, for example, is affected by loss of Opa would be of interest, but doing more ChIP is beyond the scope of the current study (and not possible due to current lab shutdown). We hope that the example above and additional data presented in the manuscript, is deemed satisfactory.

The prior analysis of Runt binding to the sog distal enhancer provides a mechanism for them to directly test the requirement for Opa-mediated accessibility in facilitating Runt binding. ChIP-qPCR for Runt on the sog distal enhancer reporter with the mutated Opa binding sites (Figure 1) and/or in the shRNAi knockdown would begin to address whether Opa has these additional defining features of a pioneer factor. In addition, the global effect of Opa-mediated accessibility on gene expression could be analyzed by RNA-sequencing.

It would be interesting to determine if Run binding is affected by Opa, but we think this analysis is beyond the scope of the current study. We aim to follow up this idea in a future study.

However, we did analyze Opa’s effect on gene expression through RNA-seq. Single embryo RNA-sequencing experiments, performed on nc14D *sh_opa* and nc14D control samples, are shown in new Figure 3, and a list of the statistically significant down-regulated (as well as up-regulated) genes in *sh_opa* are presented in Figure 3—source data 1. The text was revised to include these new data.

5) There are a number of citations that should be added to the manuscript.Kwasnieski et al., 2019 should be included with Ali-Murthy and Lott.

Added.

Xu et al., 2014 and Yanez-Cuna et al., 2014 should be included with Harrison, Liang and Nien.

Added (however, we believe that the reviewer intended to suggest Yanez-Cuna et al., 2012, which was added in addition to Xu et al).

Sun et al., 2015 should be included with Schulz.

Added .

Some mention of Li et al., 2008 and the included demonstration that A/P factors bind D/V enhancers should be included.

Added.

Papers from which ChIP-seq data sets were analyzed should be cited.

We have added an additional reference to Li et al., 2014 in this particular position, as suggested by the reviewer.

Harrison et al., 2010 should be substituted for Schulz et al., 2015.

Changed.

Reviewer #2:General comments:The title is “Odd-paired is a late acting pioneer factor…”. The defining properties of a pioneer factor are: a) protein binding to nucleosomal DNA; b) retention of the protein during mitosis; c) a general requirement for establishing/maintaining accessibility of the genome; d) cis-regulatory element binding prior to target gene activation; and e) a general function in reprogramming. While not all of these properties strictly need to be met to declare a TF a pioneer factor, the current manuscript only demonstrates that opa is necessary for accessibility. The title requires nuance, and the authors should discuss similarities and differences between opa and other pioneer factors within the body of the manuscript.

As mentioned above in response to reviewer #1 comment 4, we understand the concern. We have chosen to refer to Opa as a “pioneer-like” factor in the title, but also did add additional data to support the view that it functions as a pioneer. New paired-end ATAC-seq data were added, which allowed identification of nucleosome signatures; these data were added to Figure 5E,F and supports the view that Opa establishes accessibility by accessing previously nucleosome-bound regions.

The claim that opa acts as a timing factor is not fully supported by the data. Zelda-mediated activation timing has been demonstrated by accelerating activation with extra Zld sites or delaying it via deletion of Zld sites (Foo et al., Crocker et al., Dufourt et al., Yamada et al., etc.). To support opa regulation of timing, similar experiments would strengthen this claim immensely. Alternatively, the authors could drive maternal expression of opa and examine the change in temporal behavior on their existing Sog-MS2 transgene.

Because (i) *opa* is expressed in nc14 and acts together with Zld (which works even earlier) as well as independently to support zygotic gene expression broadly throughout embryos and (ii) can affect the accessibility and expression of enhancers in nc14, we feel that referring to Opa as a timing factor is supported. Furthermore, we provide case examples that support the view that Opa functions as a timing factor. For example, in Figure 6D in situ data for *hb*, *hb* expression pattern is delayed in *opa* mutants (i.e. expression of hb_shadow enhancer perdures at nc14B, but yet the hb_stripe is delayed). Moreover, the new ATAC-seq data from nc14B *opa* ectopic expression (UAS-*opa*) suggests chromatin opening occurs at an earlier time, whereas that for nc14D *sh_opa* embryos suggests a loss of chromatin accessibility relative to equivalently staged wildtype.

Examining whether maternal expression of Opa or addition of Opa binding sites results in changes to the temporal behavior of the sog-MS2 transgenes would be of interest, but we do not have these data and hope to include this type of analysis in a future publication looking more closely at Opa mechanism of action. We hope that the broad array of data presented here currently: live imaging of MS2 reporter with deletion of Opa binding sites, whole genome ChIP-seq, RNA-seq, and ATAC-seq data – including nucleosome displacement analysis- is satisfactory for a first paper on this pioneer-like factor.

Specific comments:1) Analysis of opa regulation of SogD-MS2 transgene– In Figure 1A, a hole could be indicative of an unhealthy embryo. Moreover, Video 1 presented in the supplementary data does not correspond to these still images. The authors should provide images from a healthy embryo and show the corresponding video.

Screenshots in Figure 1A-C have been replaced with data from the imaging of another embryo; the embryo that also corresponds to Video 1.

– In Figure 1, the number of videos should be indicated. Their quantification in terms of % of activation should be moved from the supplementary data to the main figure.

The number of videos per genotype is included in the figure legend and the quantification was moved to the main figure (see Figure 1B), as suggested. However, we retained our standard quantification that provides spatial information across the embryo width (DV axis).

– In Figure 1—figure supplement 1, the authors should add the corresponding sog_unmutated control panels and the quantification of each genotype in terms of % activation.

Control panels, as well as quantification data for these have been added using our standard quantification approach.

– The images in Figure 1B suggest that reporter expression abnormally persists in the mesoderm of sogD_ΔOpa at nc14a. If true, the authors should comment on this result. Did the authors check that the opa mutations do not affect twi or sna binding sites?

We have no evidence of abnormal ventral *sog* expression in the *sogD_Δopa* mutants; quantification of the raw data did not reveal any significant increase in the % of active nuclei present in ventral regions between *sogD_Δopa* and the control (see Figure 1B, bottom). Similarly, *sog* expression is not expanded in *opa1* mutant embryos (see Figure 1—figure supplement 1D). Because embryos rotate, the y-axis represents % egg-width (EW) such that the width of stripes can be compared but does not correlate with actual DV position (as described in Koromila and Stathopoulos, 2019 that details our quantitative analysis).

Additional screenshots from another independent video of *sogD_Δopa* that shows more clearly that expression is repressed in ventral regions was added to Figure 1—figure supplement 1. Unfortunately, this and other videos that clearly show repression were excluded from our quantitative analysis, because they did not meet other criteria (i.e. healthy embryo, lateral view of expression pattern, and length of video).

Lastly, the mutagenesis of Opa binding sites within the *sog_Distal* enhancer did not affect any Twi, Dl or Sna binding sites, as far as we could tell (i.e. no effect on sequences matching to consensus binding sites for these factors).

2) ChIP-seq comparison of Zelda vs. opa– The reference for the Zelda ChIP-seq data should be indicated (subsection “Assay of overrepresented sites associated with Opa ChIP-seq peaks”). In particular, are opa ChIP 3h samples compared to an equivalent Zelda 3h dataset (and opa 4h to Zelda 4h)?

Reference to Harrrison et al., 2011 was added at this position. Embryos used for Opa ChIP were a one hour time window collection centered at 3 h (2.5-3.5 hr), which encompasses nc14, but the Opa (4h) sample (3.54.5h) likely does not encompass nc14 (or it is greatly underrepresented). We compared only our Opa (3h) ChIP-seq sample to both Zelda nc13-nc14 and Zelda nc14 late ChIP-seq samples, and the results did not change much. Regarding the data presented in Figure 2, displaying a comparison of Zelda nc13-14 to Opa 3h, all the data was very similar when we compared Zelda nc14 late to Opa 3h except that we lost the Cad site enrichment from the Zld_only regions. To reduce the complexity, we only refer to the Zelda nc13-14/Opa 3h comparison, as it was comprehensive and the co-enrichment of Cad and Zld sites might be of interest to the field.

3) Investigating the role of opa vs. Zelda for chromatin accessibility– The authors did not justify the need for single embryo ATAC-seq. Single embryo studies are most useful if the exact developmental timing is known. If the authors did precisely time their experiment, this value should be reported with the data.

We apologize for being unclear, as we definitely did precisely time the single embryos that were used for ATAC-seq analyses. Additional information regarding the stage of the embryos has been included in the updated figures and the main text; details about our collection approach was added to the Materials and methods. Specifically, single *sh_opa* and control embryos were assayed at nc_14D (nc14+45’), whereas UAS-opa and control embryos were assayed at nc14B (nc14+20’). Embryos were collected at 26°C, and moved to our microscope room (23°C); developmental progression was viewed under a microscope and embryos harvested at the appropriate timepoints (i.e. nc14+45min and nc14+20, respectively).

– The authors should explicitly state that they used the same Gal4 driver for RNAi Zelda and RNAi opa. (subsection “Global changes in chromatin accessibility result upon knock-down of Opa”)

The main text was updated accordingly. We used the same MTD-Gal4 driver (Petrella et al., 2007) in both cases. Furthermore, we also now note that this approach, using the sh Zld RNAi construct with MTD-Gal4, was used previously for Zld RNAi by Sun et al., 2015.

– To underline that opa-bound enhancers are active later than Zelda-only enhancers, the authors should show RNA-seq tracks for the genes exemplified in Figure 3 A-H

A list of the RNA-seq values is provided in Figure 3—source data 1. For a subset of genes, these data are highlighted in Figure 3D. Because many genes expressed in the early embryo are controlled by multiple enhancers and Opa may affect only a subset, a loss of opa does not always lead to a large effect on total RNA expression levels for a gene.

– The authors mention that the opa1 mutant phenotype is comparable to that of opa RNAi embryos, but this is not shown or cited from another publication. Figure Sup4 should be complemented by similar FISH/immunolabeling in opa RNAi embryos.

To address phenotypic comparison of *opa1* and *sh_opa* mutants, we added side-by-side comparison of *sog* and *en* staining and moved these data to a main figure (i.e. Figure 3B).

– The RNAi stock used to deplete opa is not homozygous viable (given the stock described in the Materials and methods), possibly suggesting off-target effects of the RNAi. To circumvent this possibility, the authors should perform single embryo ATAC-seq on opa1 mutants. If the accessibility results are similarly affected to those in opa RNAi then it would strengthen their conclusions.

According to Hu et al., 2013, the *sh_opa* construct HMS01185 has zero off targets. Nevertheless, single embryo *opa1* nc14D ATAC-seq was performed, and the data were compared with *sh_opa* – shown in Figure 5C,D,G. Differential peak analysis showed that, from the 3784 control peaks that were not accessible in *sh_opa,* 91.1% (3448 peaks) overlap with the peaks that were not accessible in *opa1* mutants vs. control at nc14D (Figure 5—figure supplement 1A). Unfortunately, a more extended computational analysis using *opa1* was not possible due to low mapping rate (for unknown reasons); we are lacking a high quality *opa1* replicate sample. Due to the pandemic, the lab was shut down and we were not able to produce these additional data with opa1, but did add a statement about zero off targets for *sh_opa*. If anything, we are missing additional Opa effects by assaying the *sh_opa* phenotype.

– The author seems to have performed new ATAC-seq experiments on RNAi Zld, but the driver employed needs to be specified. Could the authors compare their results with published single embryo ATAC-seq in Zelda mutants (Hannon, eLife, 2017)?

We used the same MTD.Gal4 driver for RNAi knockdown of both *zelda* and *opa*.

We used nc14D embryos for *sh_zld* ATAC-seq experiments, and also used single-end sequencing for this particular experiment. Hannon et al. used nc14B embryos for zld mutant embryo ATAC-seq and paired-end sequencing. We had intended to repeat the *zld* ATAC-seq sequencing using paired-end (as we did for *opa* samples), but due to the lab shut-down because of the pandemic – we were not able to. For these reasons (i.e. difference in stage and sequencing method), our experiment and those in the other study are not directly comparable. Our *sh_zld* experiments show different trends than *sh_opa*; as MTD-Gal4 is common to both (as well as to the UAS-opa mediated ectopic expression), we do not believe MTD-Gal4 is responsible for the accessibility changes we observe (which are opposite in the case of Opa: Figure 5—figure supplement 1, and vary between *opa* and *zld* knockdowns).

– To be rigorous, the authors should have used a RNAi-white crossed with the same MTD-gal4 line as a control and not WT embryos. However, I think this experiment is less important than performing ATAC-seq in opa1 mutants.

ATAC-seq in single *opa1* mutant embryos has been performed and presented in Figure 4—figure supplement 2. In addition, we compared *opa1* mutant and *sh_opa* (i.e. *opa* RNAi using short hairpin *opa* construct x MTD-gal4) embryos, as well as compared *sh_opa* RNAi to MTD-Gal4, UAS opa mediated-ectopic expression (“UASopa”): There is 88.5% overlap (3301 peaks) between the *opa1* and *sh_opa* closed chromatin peaks (versus open peaks in control), and 75% overlap (3187 peaks) between the *sh_opa* closed (accessible/open peaks in control) and *UAS opa* open peaks (non-accessible/closed peaks in control). See Figure 5—figure supplement 1A. The concordance of *opa1* and *sh_opa* phenotypes, and the opposite effect of *sh_opa* and UAS-*opa* support the view that the accessibility changes we detect are Opa-dependent.

As discussed in the comment above, unfortunately, while the experiments above were all sequenced using pair-end sequencing (repeat experiments conducted during the revision period), we were not able to repeat the sh zld experiments and as they were sequencing using single-end sequencing these sets were not comparable in large statistical tests.

However, we do detect different accessibility in *sh_opa* versus sh zld experiments; as both these crosses contained MTD-gal4, and *opa1* mutant and *sh_opa* results are concordant, it is unlikely that the gal4 alone grossly affected accessibility.

4) opa and chromatin accessibility– Figure 4D suggests that accessibility seems much higher for Zelda/Opa common targets than for each TF target independently. Accessibility seems also higher for opa peaks compared to Zelda peaks. Is this data quantitative or semi-quantitative at all, and is there a way to explain that within the text? Can these curves be statistically compared? Can the authors produce similar graph at earlier and later stages?

Statistics of accessibility have been provided at the reviewer's request (see Figure 5—figure supplement 1A: Venn diagram comparing the number of closed chromatin peaks in *sh_opa* (3784), *opa1* (3448) versus control at nc14D, and more open chromatin peaks in *UAS opa* versus control at nc14B). However, we decided to delete original Figure 4D, since it did not add additional meaningful biological information to our study, especially after the addition of all the new data, in particular the *sh_opa* RNA-seq (Figure 3C-E) that allowed us to focus chromatin accessibility analysis on a subset of genes that exhibit opa-dependent gene expression changes (Figure 6A,B).

– The accessibility results of Figure 4 should be complemented with Zelda and opa ChIP-seq tracks and organized to better emphasize the 3 groups of accessibility identified by the authors.

Figure 4 was updated accordingly to show these suggested data, and currently displayed in Figure 5—figure supplement 1 B,C and E,F.

– It would be interesting to compare the opa peaks that are accessible independently of Zld to the genes that remain accessible in Zld mutants (Harrison, 2010, Schultz, 2015, Hannon, 2017).

This is a good idea, but we feel that more detailed comparisons between Opa and Zld action are best left for a future study that would be supported by enhancer analysis/mutagenesis etc.

I found Figures 4 and 5 to be confusing. If the central idea of Figure 4 is to show that opa is responsible for chromatin accessibility, the authors should only present WT vs. opa RNAi. Then in Figure 5, they could add the Zelda RNAi comparison.

We appreciate the feedback, and have added new Figure 3 with new *opa sh RNAi (sh_opa)* data. We have worked to improve the flow, but note we did not add Zld RNAi to the main figure.

– Subsection “Opa-only occupied peaks require Opa to support their accessibility at mid-nc14”/Figure 4G: the tracks give no information on whether Zelda binds the eve LE enhancer. The authors could add the ChIP-seq tracks as performed in Figure 4—figure supplement 2A-B to clarify this. Additionally, the loss of accessibility in Zelda RNAi at this enhancer is shown but not commented on in the text.

In the original figures, information on Zelda binding at the eve LE enhancer was provided in Figure 3B. Figure 4 has been updated and the information is now included in Figure 4B. A few sentences were also added in the main text regarding the co-binding by Zld and the expected loss of accessibility, as suggested by the reviewer.

– Also the panels of figures mentioned in the text are confusing and need revising. For example: Figure 3 is mentioned even though it does not show any accessibility data. Figure 4—figure supplement 1F does not exist.

Thank you for bringing this point to our attention. All the figures references were checked and revised to promote clarity wherever we deemed it necessary.

In particular, Figure 3 was mentioned because we were referring to Opa-bound regions. However, we understand that our figure calls were confusing, and we have worked to promote clarity in the revised manuscript.

– Subsection “Opa-only occupied peaks require Opa to support their accessibility at mid-nc14” paragraph three: The authors use the term opa1 mutant, when in fact they are looking at opa RNAi-mediated transcript depletion, which is significantly different.

Thank you for bringing this error to our attention. “Opa mutants” has been replaced with “*opa RNAi*” in all three of the places in the text appended below.

“…supporting the view that Zld is pivotal for early expression (Figure 6C). In contrast, in *opa* mutants, at nc13, little difference was observed; *opa* is not expressed at nc13, therefore mutants would not be expected to affect patterning at this stage (Figure 6C). Later, at nc14B, the *opa* mutants do exhibit expression defects. *sog* is diminished in *opa* mutant relative to wildtype (Figure 6D); and the hb pattern found in *opa* mutants…”

Reviewer #3:I do have some concerns about the staging of the embryos. In particular, the central claim of the paper is that Opa coordinates with Zelda in regulating gene expression, because it binds to many of the same genomic regions, and for some of these regions, opa knockdown and zld knockdown both affect accessibility. Simultaneous Opa+Zld binding is inferred by comparing Opa ChIP-seq peaks to a published Zld ChIP-seq dataset. However, the Zld dataset uses embryonic stages (nc13 and early nc14) from before Opa is expressed, meaning that the possibilities of simultaneous binding vs. sequential binding cannot be distinguished.

Based on our Opa immuno-staining data, Opa protein is first expressed (detectable) at nc14B. The Zld ChIP-seq “early” data were taken from nc13-nc14 embryos, suggesting that there would be an overlap between Opa and Zld at nc14B, which is considered “early nc14”. In any case, we also compared our Opa 3h ChIP-seq dataset with the late nc14 Zld ChIP-seq dataset (see next point).

If possible, I would have the authors re-run their analysis using the "late nc14" dataset from the same paper instead, which seems a more appropriate comparison for their purposes. As an optional extension, explicit comparison between the early and late Zld datasets could also give interesting hints as to the nature of any Opa/Zld interaction – for example when Opa starts binding to the genome in late nc14, does this cause Zld binding at these loci to increase or reduce, relative to other Zld-bound loci where Opa is absent?

The " nc14 late" Zld ChIP-seq (GSM763062) dataset was also analyzed and new figures have been generated and updated accordingly.

[Editors’ note: what follows is the authors’ response to the second round of review.]

Reviewer #2:This is a much improved manuscript that has worked to address many of the significant concerns from the prior submission. The additional data strengthen the conclusions of the manuscript. Furthermore, the focus on the Opa (3h) data successfully streamline the manuscript such that the focus remains on the conclusions most robustly supported by the data. We have only minor issues that should be addressed prior to publication.1) The additional data with the ATAC-seq upon precocious expression of Opa and the analysis on nucleosome distance reported in Figure 5 are nearly identical to experiments published by Soluri et al., 2020. As such, this publication must be briefly discussed and cited. It is gratifying to show that these results are robust across laboratories and acknowledgement of this fact does not decrease the impact of this publication. We feel that this addition to the manuscript is necessary for acceptance.

We ectopically expressed *opa* and looked for changes in accessibility, whereas Soluri et al. looked at accessibility changes in *opa* mutants. These are opposite experiments. For that reason, our analytical results shouldn't be expected to be the same as those of Soluri et al. paper. It is consistent to the model that increasing levels of Opa leads to longer distances between the nearest nucleosome and the Opa-bound position because of increased nucleosome displacement. We added reference to this other study as well as a sentence describing their experiment as we are in agreement with the reviewer that it strengthens the view that Opa affects nucleosome positioning.

2) Figure 3D is confusing as it appears that only a very small handful of genes are bound by Opa. This is obviously not the case as shown in Figure 3—figure supplement 1F, but the authors should consider a different way of highlighting specific genes. As it is, the black dots are labelled "Opa only" but these are clearly only a small fraction of the Opa only bound genes in this volcano plot. Similarly for the yellow Opa/Zld genes. It could be useful to report on the plot the % of down- and up-regulated genes that are proximal to an Opa binding site.

Figure 3D was modified to remove Opa only and Opa/Zld designations, which instead are listed in new Figure 3—source data 1.

3) For the various heat maps, the order in which peaks are ranked should be clearly indicated. For example, in Figure 4F it is unclear whether each heat map is ranked separately or whether one can compare across heat maps. Similarly, the method for ranking peaks should be provided for Figure 5A and B.

Order of the genomic regions (y-axis) in the heatmaps:

The first sample in the heatmap was used for sorting the genomic regions based on descending order of mean signal value per region. All other samples were plotted using the same order determined by the first sample. We added a note to the legend of Figure 4 to clarify this point, which relates to all other heatmaps presented in other figures also.

4) There are a few typos/formatting errors.- Results paragraph three, the authors state that Opa expression is reduced at nc14c for a mutant reporter. It is not clear whether the authors mean Opa expression in this case.

The text is now corrected and Opa is replaced with *sog_Distal*.

- In many of the figures, the symbol for α in antibody staining is an "a".

Figure 1 and 3 were corrected.

- The authors state that there is a "significant increase in chromatin accessibility across Ope-bound regions (Figure 5A)." While these data are compelling, the word significant implies some sort of statistical analysis. If such an analysis was performed this should be reported. Otherwise, a change in word choice should suffice.

The word “significant” is now replaced with “clear”.

- The citation (Blythe, 2016) in paragraph three of the Discussion should presumably be Blythe and Wieschaus, 2016.

Fixed.

- In the legend for Figure 1I "*Mus musculus*" should have the genus name capitalized and should be italicized.

Fixed.

- The inclusion of Su(H) in Figure 6E is confusing and should be removed.

Su(H) was removed from the figure.